# NR2F1 regulates regional progenitor dynamics in the mouse neocortex and cortical gyrification in BBSOAS patients

Michele Bertacchi[1,2] (ID), Anna Lisa Romano[1], Agnès Loubat[1], Frederic Tran Mau-Them[3,4], Marjolaine Willems[5], Laurence Faivre[3,4,6], Philippe Khau van Kien[7], Laurence Perrin[8], Françoise Devillard[9], Arthur Sorlin[3,4,6,10], Paul Kuentz[3,11] (ID), Christophe Philippe[3,4], Aurore Garde[4,6], Francesco Neri[12,13], Rossella Di Giaimo[14,15] (ID), Salvatore Oliviero[12], Silvia Cappello[15] (ID), Ludovico D'Incerti[16], Carolina Frassoni[2] & Michèle Studer[1,*] (ID)

## Abstract

The relationships between impaired cortical development and consequent malformations in neurodevelopmental disorders, as well as the genes implicated in these processes, are not fully elucidated to date. In this study, we report six novel cases of patients affected by BBSOAS (Boonstra-Bosch-Schaff optic atrophy syndrome), a newly emerging rare neurodevelopmental disorder, caused by loss-of-function mutations of the transcriptional regulator *NR2F1*. Young patients with *NR2F1* haploinsufficiency display mild to moderate intellectual disability and show reproducible polymicrogyria-like brain malformations in the parietal and occipital cortex. Using a recently established BBSOAS mouse model, we found that *Nr2f1* regionally controls long-term self-renewal of neural progenitor cells via modulation of cell cycle genes and key cortical development master genes, such as *Pax6*. In the human fetal cortex, distinct NR2F1 expression levels encompass gyri and sulci and correlate with local degrees of neurogenic activity. In addition, reduced NR2F1 levels in cerebral organoids affect neurogenesis and PAX6 expression. We propose *NR2F1* as an area-specific regulator of mouse and human brain morphology and a novel causative gene of abnormal gyrification.

**Keywords** BBSOAS; cell cycle dynamics; cortical folding; neurodevelopmental disease; *NR2F1/COUP-TFI*

**Subject Categories** Development; Neuroscience; Stem Cells & Regenerative Medicine

**The EMBO Journal (2020) 39: e104163**

## Introduction

During development, neural progenitor (NP) cells shape the cerebral cortex by means of a delicate balance between neurogenesis (differentiation) and progenitor maintenance (self-renewal) (Florio & Huttner, 2014). Different NP subtypes concur to produce neurons in a direct or indirect (via intermediate progenitor—IP) way; differentiating neurons migrate to the cortical plate (CP), where they settle down to mature into distinct neuronal subtypes radially organized into layers and circuits (Dehay & Kennedy, 2007; Kumamoto & Hanashima, 2014). However, layers are not homogenously distributed along the antero-posterior (A-P) and latero-medial (L-M) axes of the developing neocortex, generating histological discontinuities

1 Université Côte d'Azur, CNRS, Inserm, iBV, Paris, France
2 Clinical and Experimental Epileptology Unit, Fondazione IRCCS Istituto Neurologico Carlo Besta, Milano, Italy
3 UMR1231 GAD, Inserm - Université Bourgogne-Franche Comté, Dijon, France
4 Unité Fonctionnelle Innovation en Diagnostic Génomique des Maladies Rares, FHU-TRANSLAD, CHU Dijon Bourgogne, Dijon, France
5 Hôpital Arnaud de Villeneuve, Service de Génétique Médicale, CHU de Montpellier, Montpellier, France
6 Centre de Référence maladies rares « Anomalies du développement et syndromes malformatifs », Centre de Génétique, FHU-TRANSLAD, CHU Dijon Bourgogne, Dijon, France
7 Hôpital Carémeau, UF de Génétique Médicale et Cytogénétique, Centre de Compétences Anomalies du Développement et Syndromes Malformatifs, CHU de Nîmes, Nîmes, France
8 Unité Fonctionnelle de Génétique Clinique, Hôpital Robert Debré, Paris, France
9 Département de Génétique et Procréation, Hôpital Couple-Enfant, CHU de Grenoble, Grenoble, France
10 Centre de référence maladies rares « Déficiences intellectuelles de causes rares », Centre de Génétique, FHU-TRANSLAD, CHU Dijon Bourgogne, Dijon, France
11 Génétique Biologique, PCBio, Centre Hospitalier Universitaire de Besançon, Besançon, France
12 Epigenetics Unit, Italian Institute for Genomic Medicine, University of Torino, Torino, Italy
13 Leibniz Institute on Aging, Fritz Lipmann Institute (FLI), Jena, Germany
14 Department of Biology, University of Naples Federico II, Napoli, Italy
15 Max Planck Institute of Psychiatry, München, Germany
16 Neuroradiology Unit, Fondazione IRCCS Istituto Neurologico Carlo Besta, Milano, Italy
*Corresponding author. Tel: +33 489150720; E-mail: michele.studer@unice.fr

and cytoarchitecturally defined regions, which will differentiate into functional areas (Alfano & Studer, 2013). Although it is well accepted that presumptive areas, called proto-areas, get specified during early development (Cadwell et al, 2019), the underlying molecular and cellular mechanisms are still not fully elucidated. In particular, it is not well known whether abnormal area-specific development in humans can lead to particular brain malformations.

Malformations of cortical development (MCDs) represent a heterogenous group of neurodevelopmental diseases in which structural brain anomalies are associated with complex neurodevelopmental disorders (Juric-Sekhar & Hevner, 2019). They can be classified on the basis of their etiology: impairments in NP proliferation, neurogenesis, neuron migration, differentiation, apoptosis, and/or cortical organization (Manzini & Walsh, 2011; Barkovich et al, 2012). While disorders of reduced proliferation result mainly in microcephaly, altered cell migration can lead either to various types of cellular heterotopia, in which differentiating neurons fail to reach their correct destination and form aggregates in ectopic positions, or to a thick disorganized CP often leading to lissencephaly (Parrini et al, 2016). Finally, affected NP physiology and cell migration have also been hypothesized to cause abnormal gyrification, such as polymicrogyria (PMG), a heterogenous group of malformations characterized by abnormal shape and number of cortical gyri (Barkovich et al, 2012; Sun & Hevner, 2014; Parrini et al, 2016). Gyral abnormalities are often associated with mild to moderate intellectual disability (ID), infantile spasms, and impaired oromotor skills (Jansen, 2005; Parrini et al, 2016). Despite some genetic abnormalities underlying MCDs, particularly related to cell migration, are starting to be elucidated (Cappello et al, 2013; Juric-Sekhar & Hevner, 2019; Uzquiano et al, 2019), the cellular and molecular mechanisms underlying abnormal gyrification are still not well understood.

The transcriptional regulator Nr2f1 (Nuclear Receptors Nomenclature, 1999), also known as COUP-TFI, is an orphan nuclear receptor belonging to the superfamily of the steroid/thyroid hormone receptors (Qiu et al, 1994; Bertacchi et al, 2018). Expressed along the neocortex following a high-posterior to low-anterior gradient in progenitors and post-mitotic neurons, Nr2f1 plays several key roles during areal organization (Zhou et al, 2001; Armentano et al, 2007; Bertacchi et al, 2018) and controls both neurogenesis (Faedo et al, 2008) and neuronal cell migration in embryonic (Alfano et al, 2011; Lodato et al, 2011; Parisot et al, 2017) and adult stages (Bovetti et al, 2013; Flore et al, 2016; Bonzano et al, 2018). Although Nr2f1 has been extensively studied in post-mitotic sensory area identity acquisition (Alfano et al, 2014), its ultimate function in NP physiology during early corticogenesis, when cell proliferation needs to be tightly controlled, is poorly understood.

Recent reports show that Nr2f1 and NR2F1 follows a similar graded expression profile in the mouse and human brains, respectively (Alzu'bi et al, 2017a,b; Yang et al, 2017), in accordance with its evolutionary highly conserved sequence and protein homology (Bertacchi et al, 2018). In the last few years, deletion and point mutations in the NR2F1 locus have been identified in patients with optic nerve atrophy associated with developmental delay, autistic features, epilepsy, and ID (Brown et al, 2009; Cardoso et al, 2009; Al-Kateb et al, 2013; Bosch et al, 2014; Chen et al, 2016, 2017; Kaiwar et al, 2017; Martín-Hernández et al, 2018). At present, patients with NR2F1 haploinsufficiency are diagnosed for the Bosch-Boonstra-Schaaf optic atrophy syndrome (BBSOAS), an emerging neurodevelopmental autosomal dominant disorder (OMIM #615722; ORPHANET #401777) leading to

a broad range of clinical phenotypes associated with visual and cognitive deficits reviewed in Bertacchi et al (2018). Given the high prevalence (almost 70%) of visual deficits in BBSOAS patients (Bosch et al, 2014; Bertacchi et al, 2018), previous studies have mainly focused on elucidating the chiasm and optic nerve atrophy of young children. This, together with the difficulty of obtaining high-resolution magnetic resonance imaging (MRI) at young age, has limited the investigation of possible brain malformations, which could correlate with the high frequency of ID (more than 80%) and epileptic seizures (40%) in BBSOAS patients. New reports are constantly expanding the BBSOAS clinical spectrum (Chen et al, 2016; Martín-Hernández et al, 2018; Bojanek et al, 2019; Park et al, 2019; Rech et al, 2020), as more patients are identified and the pathophysiology of different brain regions or different organs starts to be explored.

In this study, we describe for the first time abnormal folding in six new patients with de novo point variants within the start codon for translation, the DNA-binding Domain (DBD) or the ligand-binding domain (LBD) of NR2F1. All these patients show ID features, behavioral disorders, and delayed motor and language development, besides optic atrophy. Interestingly, five out of six patients have a unilateral PMG-like pattern along the parieto-occipital cortical region, and two patients show abnormally elongated occipital convolutions. To understand the mechanisms at the basis of this regionalized morphological impairment, we used the mouse Nr2f1 loss-of-function model, recently established to represent a reliable animal model for BBSOA syndrome (Bertacchi et al, 2019). We particularly focused on NP physiology and showed that Nr2f1 finely controls the balance between self-renewal of distinct progenitor cell types and differentiation of specific neuronal classes in the occipital cortex. Notably, we observed cell cycle shortening and abnormally increased proliferative cell divisions in the mutant embryo, resulting in an early expansion of the NP pool. At the molecular level, we found that Nr2f1/NR2F1 influences NP physiology in mouse cortex and human organoids by modulating PAX6, a known patterning gene controlling progenitor expansion, as well as the cell cycle inhibitor P21 in the mouse cortex. Finally, we show that genetic manipulation of Pax6 levels rescues self-renewal features of Nr2f1-deficient NPs. Together, our data demonstrate that Nr2f1 operates a regional control of NP physiology, linking local progenitor activity to positional identity along neocortical axes. This can have direct consequences on brain morphology, as distinct cortical regions are abnormally developed in BBSOAS patients and in the mouse model. We propose that Nr2f1/NR2F1 orchestrates cortical size and folding in an area-specific manner by fine-tuning cell cycle progression and local production of specific progenitor subtypes. Finally, we suggest aberrant local gyrification and cortical malformations as novel morphological features expanding the BBSOAS clinical phenotype.

# Results

## NR2F1 loss-of-function variants cause cortical malformations and abnormal gyrification in BBSOAS patients

To understand the etiology of cortical malformations in human patients and their correlation with ID, we ascertained a collection of seventeen novel cases of patients (M. Bertacchi and M. Studer, unpublished) with BBSOA syndrome. Previously published high-

**Table 1. Novel NR2F1 variants and phenotypic features of patients with BBSOA syndrome.**

| Patient ID | Gender/ Age at MRI evaluation | Genetics | In Silico | Clinical features: general | Neurological exam: cognitive level | Ophthalmologic disorders | Brain MRI: general | Brain MRI: morphological malformations |
|---|---|---|---|---|---|---|---|---|
| P1 | Female, 1 year 4 months | chr5:g.92921154G>A; NM_005654.4:c.425G>A; p.Arg142His; *de novo* | GnomAD: 0 CADD: 33 Polyphen-2: 1 GERP: 3.7 | Developmental delay, behavioral disorders, hypotonia, speech difficulties, stereotype movements, infantile spasms | Intellectual disability, autistic-like traits, hyperactive behavior | Amblyopia; optic atrophy | Thinning of the corpus callosum; hypoplasia of optic nerves and optic chiasm; | Abnormally elongated occipital convolutions (right side) |
| P2 | Male, 3 years 7 months | chr5:g.92923888_92923889 delinsCT; NM_005654.4: c.729_730delinsCT; p.Gln244*; *de novo* | GnomAD: 0 CADD: 50 Polyphen-2: ND GERP: 4.5 | Developmental delay, behavioral disorders, hypotonia, bilateral inguinal hernias | Intellectual disability, fine motor and praxis disorders | Hypermetropia, strabismus | Thinning of the posterior half of the corpus callosum; ventricular asymmetry, and mild ventricular enlargement | Abnormal gyral pattern of the occipital cortex at the supramarginal and angular gyri (left side) |
| P3 | Female, 5 years 10 months | chr5 g.92920844G>T; NM_005654.4: c.115G>T; p.Glu39*; *de novo* | GnomAD: 0 CADD: 36 Polyphen-2: ND GERP: ND | Developmental delay, behavioral disorders, hypotonia, speech difficulties, epileptic seizures, precocious puberty | Intellectual disability, stereotypical movements, attention deficits | Strabismus, low visual acuity, severe bilateral optic atrophy | Thinning of the corpus callosum and optic chiasm, superior cerebellar vermis atrophy, ectopic nodular heterotopy | Abnormal gyral pattern of posterior temporal, supramarginal and angular gyri (left side) |
| P4 | Female, 3 years 5 months | chr5:g.92920731T>C; NM_005654.4:c.2T>C; p.? *de novo* | GnomAD: 0 CADD: 22 Polyphen-2: 0.02 GERP: 2.3 | Developmental delay, behavioral disorders, speech difficulties | Intellectual disability, autistic-like traits, hyperactive behavior | Thinning of the optic chiasm | Thinning of the posterior half of the corpus callosum; small asymmetry of the occipital cortex | Abnormal gyral pattern of the cortex of the supramarginal and angular gyri (left side) |
| P5 | Female, 6 years 7 months | chr5:g.92921021T>C; NM_005654.4:c.292T>C; p.Tyr98His; *de novo* | GnomAD: 0 | Developmental delay, behavioral disorders, hypotonia, stereotypical movements | Intellectual disability, autistic-like traits | Optic nerve atrophy, strabismus, hypermetropia, nystagmus | Thinning of the posterior half of the corpus callosum, hypoplasia of the optic chiasm | Abnormal gyral pattern of the cortex of the supramarginal and angular gyri (left side) |
| P6 | Male, 12 years 2 months | chr5: g.92924126_92924127delAA NM_005654.4: c.967_968delAA p.Lys323Serfs*73 *de novo* | GnomAD: 0 | Delayed motor development, speech difficulties, pectus excavatum, pes planus, scoliosis, triangular face, thin upper lip, anteverted nares, high palate, gastroesophageal reflux, severe apnea | Intellectual disability, autistic-like traits | Optic nerve atrophy, strabismus, hypermetropia, low visual acuity, tubular vision | Short corpus callosum (<3rd percentile), thin optic nerves and chiasm, hypoplastic olfactory lobes | Abnormal gyral pattern of the cortex of the supramarginal and angular gyri (left side) |

resolution MRI scans have mainly focused on the optic nerve/optic chiasm hypotrophy (Bosch *et al*, 2014; Chen *et al*, 2016), resulting in poor characterization of possible brain malformations due to *NR2F1* pathogenic variants, despite the high frequency of these patients in being diagnosed with ID. In this study, we report MRI brain scans of six novel patients carrying *de novo NR2F1* variants localized along the whole gene sequence and characterized by developmental delay, behavioral disorders, speech difficulties, and autistic-like traits (Table 1; Fig 1A–G). Analysis from the MRI data revealed a thinning of the corpus callosum and optic nerve chiasm together with optic nerve hypotrophy (Fig EV1A–F), key diagnostic morphological features of BBSOAS patients (Bosch *et al*, 2014; Chen *et al*, 2016). Interestingly, careful analysis of axial images revealed subtle cortical malformations in five patients, such as altered patterns of gyrification, resembling PMG, at the level of the supra-marginal and angular convolutions of the inferior parietal cortex (Fig 1B–F). This microgyria-like abnormality was found only and always on the left side (Fig 1B–F). MRI data also disclosed an elongation of the right superior occipital gyrus without evident gyrification defects in patient (P) 1 (Fig 1A) and abnormally prominent occipital convolutions in P2 (Fig EV1B).

Genomic DNA samples from all patients were subjected to exome sequencing (ES), as previously described (Orphanomix Physicians' Group *et al*, 2018). Genome Aggregation Database (GnomAD) metrics (misZ: 4.17; pLI: 0.99) predict the *NR2F1* gene to be highly intolerant to missense mutations, in agreement with the pathogenic phenotype caused by loss-of-function (LOF) variants (Bosch *et al*, 2014; Chen *et al*, 2016). We identified six novel mutations in the *NR2F1* gene (Table 1; Fig 1G), absent from the GnomAD database. In P1, a missense variant (c.425G>A; p.Arg142His) was detected in the functional C4-type zinc-finger domain of the DBD. Several pathogenic variants have been previously described in this domain (Chen *et al*, 2016). Differently, P2 showed a stop-gain variant in exon 2 (c.729_730delinsCT; p.Gln244*) predicted to lead to nonsense-mediated *mRNA* decay, since located at 50–55 nucleotides upstream of the last intron–exon junction (Lykke-Andersen & Jensen, 2015). A novel variant, 115 bases downstream of the initiation site and leading to a truncation (c.115G>T; p.Glu39*), was identified in P3, while in the case of P4 a variant at the initiation site (c.2T>C; p?) is anticipated to interrupt protein translation. A novel missense variant was also characterized in P5, located in the functional DBD zinc-finger domain (c.292T>C; p.Tyr98His), similarly to P1. Finally, P6 carries a missense variant, located in the LBD proximal to C-terminal side and leading to protein truncation (c.967_968delAA; p.Lys323Serfs*73). In all patients, Sanger sequencing confirmed the *de novo* occurrence. No additional single nucleotide and gene copy number variant that could explain the phenotype was identified. All these patients depicted a reproducible and regional brain malformation defect, apart from P1, with very similar clinical features. Taken together, we introduced six new patients with *NR2F1* variants located at the initiation site, in the DBD and LBD, and showing similar clinical and brain malformation defects.

## NR2F1 displays distinct expression gradients along the human brain axes and micro-modules encompassing primary convolutions

The presence of local alterations in cortical folding of this new cohort of BBSOAS patients suggests that NR2F1 might act in a

regionalized manner during the gyrification process. To this purpose, we investigated NR2F1 expression pattern in different cortical regions of human gestational week (GW) 11 and GW14 fetal sections. NR2F1 expression in apical and basal NP cells as well as differentiated neurons showed a clear latero-posterior high to medio-anterior low expression gradient (Figs 1H and I, and EV1G–J), which reminds the same graded expression profile along the A-P and L-M brain axes previously described in mouse embryos (Armentano *et al*, 2006, 2007). Together with the high protein similarities between rodents and humans, these data support highly conserved expression patterns, and possibly function during evolution (Alzu'bi *et al*, 2017a,b; Bertacchi *et al*, 2018).

The human cortex differs from that of the mouse for its gyrencephalic pattern, i.e., the formation of gyri and sulci that allows an expansion of the total cortical surface (Sun & Hevner, 2014; Fernández *et al*, 2016). The first neocortical folds (also called primary convolutions), such as the Sylvian or calcarine fissures as well as the parieto-occipital sulcus, arise as early as GW9 (Kostovic & Vasung, 2009; Sarnat & Flores-Sarnat, 2016), but are more evident between GW13 and GW15, when they display a cytoarchitectural organization of progenitors that reminds that of later secondary and tertiary gyri/sulci (Kostovic & Vasung, 2009; Hansen *et al*, 2010; Zhang *et al*, 2013; Vasung *et al*, 2016; Kostović *et al*, 2019). Basal radial glia (bRG), a specialized progenitor cell type able to produce a high number of neurons and abundantly represented in gyrencephalic species at the outer subventricular zone (oSVZ) (Hansen *et al*, 2010; Reillo *et al*, 2011; Pollen *et al*, 2015), possibly contribute to cortical folding in association with other progenitor parameters (Garcia-Moreno *et al*, 2012; Kelava *et al*, 2012). The modular expression of some genes has been proved to control local abundance of neurogenic bRG cells, ultimately positioning gyri and sulci (de Juan Romero *et al*, 2015). We found that NR2F1 expression levels are finely modulated around these developing primary folds (Fig 1J–Q). While low NR2F1 levels are observed at prospective sulci (Fig 1J, K', N and P'), where SOX2$^+$ bRG and TBR2 + intermediate progenitor (IP) cells are very few (Fig EV1K–L''), NR2F1 levels increase in folded regions (Fig 1J, L' and Q), where a high amount of IP and bRG cells are formed together with increased neuronal production and tangential dispersion (Figs 1P'' and R, and EV1K–L''). Hence, micro-modules of low to medium/high NR2F1 levels are associated with distinct local distribution of progenitor types (Fig EV1L–L'') and specific abundance of cells expressing HOPX (Fig 1P–R), a *bona fide* marker of human bRG cells (Pollen *et al*, 2015). These data suggest a modular and fine regulation of *NR2F1* expression levels around forming convolutions that could be associated with the process of cortical gyrification, as previously described for other genes (de Juan Romero *et al*, 2015) (Fig EV1M).

## Nr2f1-mediated control of self-renewal potential in mouse neurospheres

The high NR2F1 expression profile in NP cells and the finding of microgyral malformations in BBSOAS patients prompted us to investigate NR2F1 in NP physiology. In mouse, Nr2f1 has been previously shown to coordinate neurogenesis (Faedo *et al*, 2008); however, its exact function within the different NP populations and in cell cycle regulation has not been investigated so far. Hence, as a

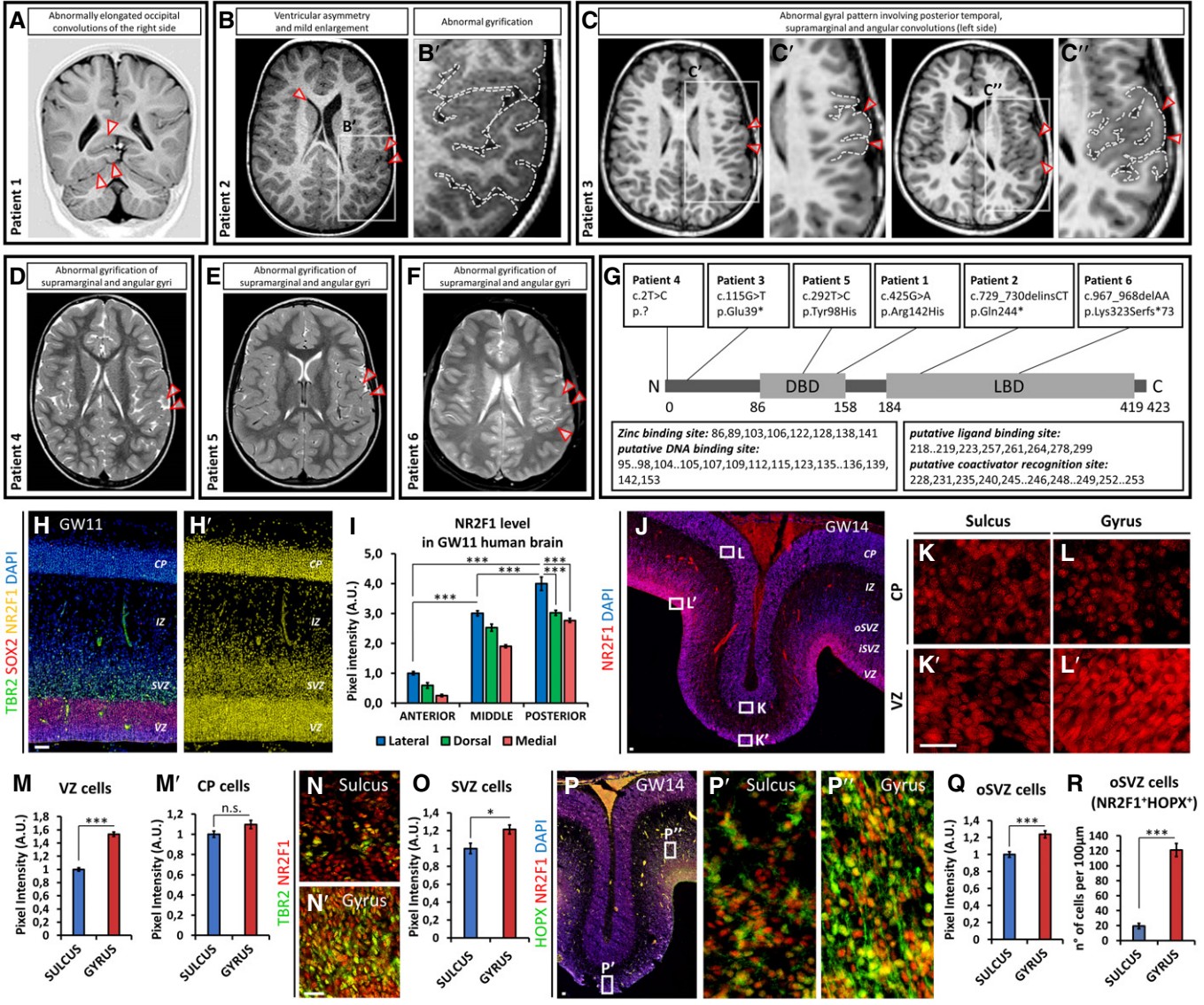

**Figure 1. Cortical malformations in BBSOAS patients with NR2F1 haploinsufficiency.**

A–F    Cortical malformations observed in six novel BBSOAS patients (See Fig EV1A–F for additional MRI panel and Table 1 for clinical features). Patient (P)1 shows abnormally elongated occipital convolutions (arrowheads in A), whereas abnormal gyrification of supramarginal and angular gyri is present in five out of six individuals (P2-P6; B-F). B′, C′, and C″ highlight the abnormally convoluted regions boxed in (B and C).

G    Schematic representation of the human NR2F1 protein sequence (based on the current annotation: https://www.ncbi.nlm.nih.gov/protein/NP_005645.1), showing novel variants of a new BBSOAS patient cohort. Key amino acids for the functioning of the DNA-binding domain (DBD) or the ligand-binding domain (LBD) are listed in boxes.

H, H′    TBR2 (green in H), SOX2 (red in H), and NR2F1 (yellow in H′) immunofluorescence (IF) of a GW11 section of human neocortex (see also Fig EV1G–G‴).

I    NR2F1 expression (quantified by pixel intensity) at different L-M levels (see also Fig EV1G–G‴) and in different regions along the A-P extent of the cortex, as indicated. $n \geq 3$ sections from $n = 1$ fetal brain.

J–L′    NR2F1 (red) IF of human GW14 neocortex, showing expression levels in the CP (K, L) and VZ (K′, L′) around primary convolutions in the posterior-most cortex (see also Fig EV1H). High NR2F1 is detected in the progenitor area of a gyrus (L′).

M, M′    NR2F1 level as measured on single cells in the VZ (M) and CP (M′) of sulci and gyri, as in (J-L′). $n \geq 4$ sections from $n = 1$ fetal brain.

N, O    NR2F1 (red) and TBR2 (green) immunostaining of the same regions shown in (J-L′). NR2F1 level as measured in single TBR2⁺ IPs by pixel intensity is shown in graph (O). $n \geq 4$ convolutions from $n = 1$ fetal brain ($n = 2$ sections).

P–R    NR2F1 (red) and HOPX (green) immunostaining of the same regions shown in (J-L′) (see also Fig EV1H). Virtually all HOPX⁺ bRGs are also NR2F1⁺ (magnification in P″) and their number is greatly increased in gyri (P″) compared to sulci (P′), as quantified in (R). At a single cell level, NR2F1 pixel intensity is higher in oSVZ HOPX⁺ cells in gyri compared to sulci (Q). $n \geq 4$ sections from $n = 1$ fetal brain.

Data information: Nuclei (blue) were stained with DAPI. In (I, R), the number of positive cells was quantified in 100 μm-width boxes, randomly placed across the cortex. In graphs, data are represented as means ± SEM. Two-way ANOVA (I) and Student's $t$-test (M, M′, O, Q, R) (*$P < 0.05$, ***$P < 0.001$). Scale bars: 50 μm. CP: cortical plate; iSVZ: inner subventricular zone; IZ: intermediate zone; oSVZ: outer subventricular zone; SVZ: subventricular zone; VZ: ventricular zone.

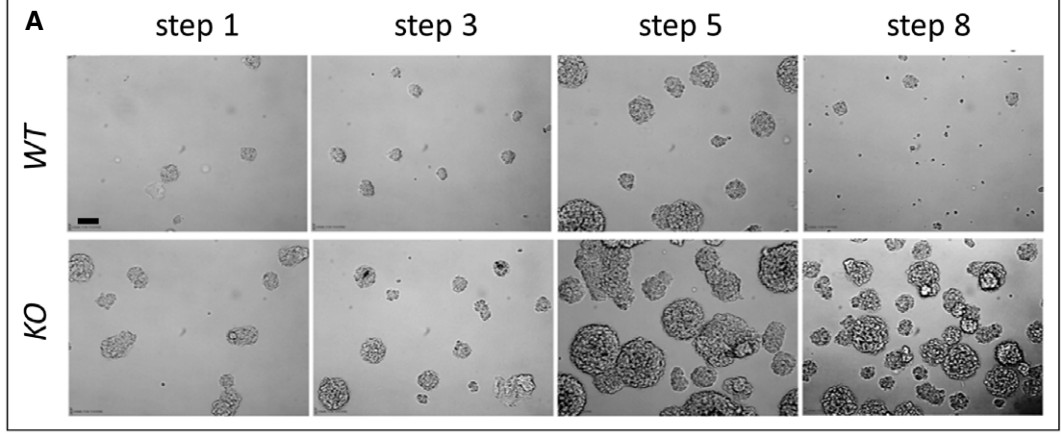

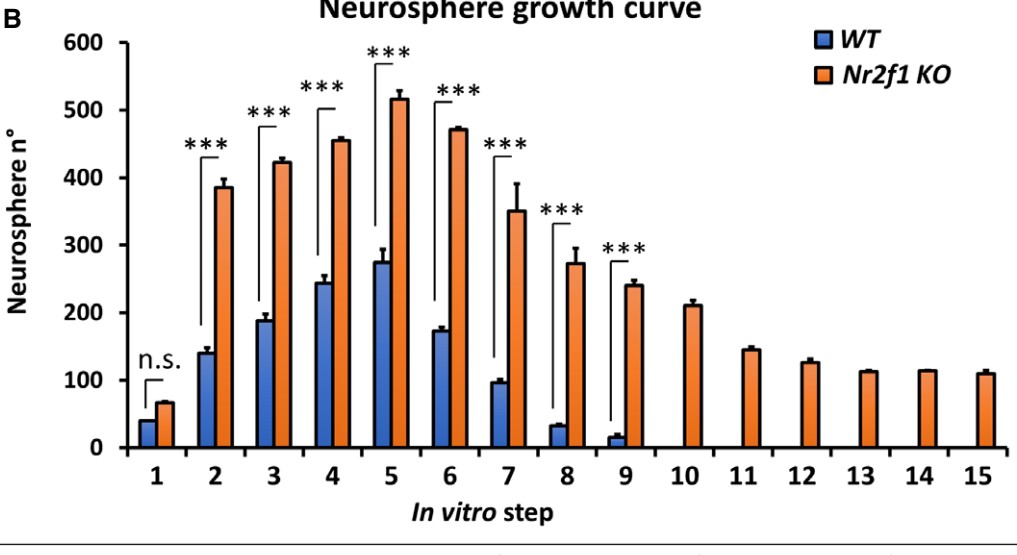

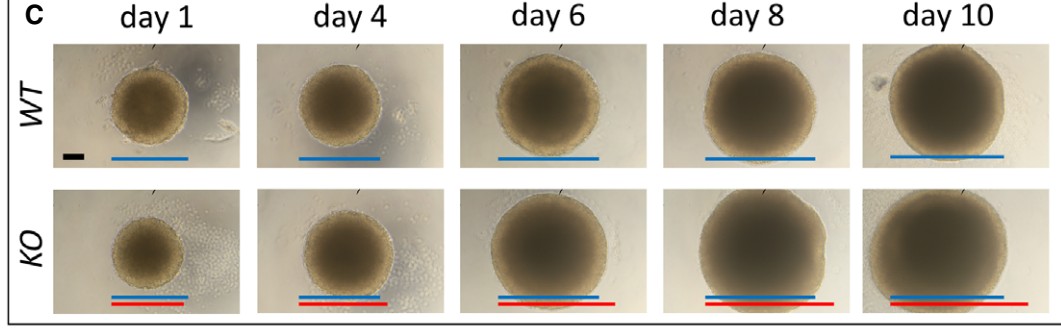

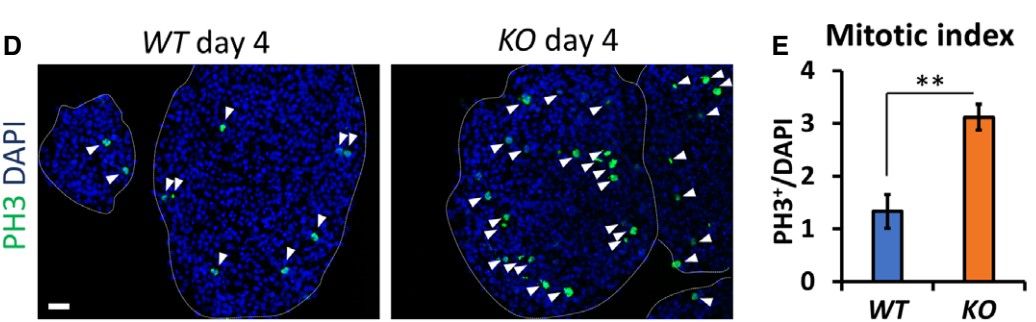

**Figure 2.**

**Figure 2. Nr2f1 regulates neural progenitor self-renewal in a neurosphere assay.**

A   Representative images of wild-type (*WT*) and mutant (*KO*) neurospheres obtained from E15.5 neocortices and cultured *in vitro*. The culture passage ("step") corresponds to 3 days of culture after dissociation and re-plating. *KO* aggregates are more numerous and larger in size.

B   Graph showing the number of neurospheres per P100 cell plate at different steps, as indicated. *Nr2f1 KO* neurospheres proliferate for a longer time (up to 30 passages tested), whereas *WT* cells exhaust around step 9. $n \geq 3$ culture plates from $n = 2$ batches.

C   Panel of *WT* and *KO* isolated neurospheres grown from day 1 to day 10. Blue and red lines represent the diameter of *WT* and *KO* neurospheres, respectively.

D, E   PH3 (green; dividing cells) IF on cross-sections of day 4 neurospheres. White arrowheads point at mitotic figures, which are quantified in (E) as percentage of dividing cells over DAPI-stained nuclei (blue). $n \geq 3$ neurospheres from $n = 2$ batches.

Data information: In (B, E), data are represented as means ± SEM. Two-way ANOVA (B) and Student's *t*-test (E) (**$P < 0.01$, ***$P < 0.001$). Scale bars: 50 μm. PH3: Phosphohistone H3.

first step in assessing *Nr2f1* action in NP self-renewal ability, we cultured NP-derived neurospheres, representing free-floating progenitor clusters, established from E15.5 *Ctrl* (*WT*) and *Nr2f1null* (*KO*) cortices (Armentano *et al*, 2006), and cultured for several passages in FGF2 and EGF-containing medium (Reynolds *et al*, 1993; Brewer & Torricelli, 2007). Interestingly, while *WT* neurospheres expanded until step 5 and underwent exhaustion by step 9 (Fig 2A and B), *Nr2f1 KO* neurospheres displayed long-term self-renewal for more than 15 steps (up to 30 passages tested; Fig 2B). Besides their capacity of continuously self-amplifying, *Nr2f1* mutant neurospheres were both more numerous and bigger in size (Fig 2A); however, we reasoned that the latter parameter could be influenced by fusion of adjacent neurospheres. Thus, we dissociated some cell clusters and isolated single neurospheres to evaluate their independent growth (Fig 2C): by day 6 of *in vitro* culture, single mutant neurospheres depicted a significantly bigger size compared to *WT* ones (10.2 ± 2.4% and 14.7 ± 1.7% diameter increase at day 6 and day 10, respectively; $P = 0.002$ and $P = 0.003$). This enhanced growth was associated with a doubled number of PH3$^+$ dividing NPs within the neurospheres (Fig 2D and E). Enhanced mitotic activity was not dependent on the specific culture protocol used, as we detected an increased number of PH3$^+$ cells also when NPs were cultured as a monolayer (Appendix Fig S1A–C). These results show that NPs have increased self-renewal capability upon *Nr2f1* loss, further reinforced by enhanced proliferation, and suggest that, normally, Nr2f1 might restrain self-renewal potential of cortical NP cells.

## Progenitor pool amplification in *Nr2f1 KO* mutants leads to occipital neocortex expansion

To dissect the role of Nr2f1 in NP cell behavior *in vivo*, we directly quantified the number of proliferating progenitors in the neocortex of *WT* and *Nr2f1 KO* at different ages by using the proliferation marker Ki67 and focusing on the lateral pallium (LP) of posterior cortices, a region of highest *Nr2f1* expression from E9.5-E10.5 (Tomassy *et al*, 2010). Ki67$^+$ proliferating NPs increased in the mutant starting from E14.5 and were maintained in high numbers until P0 compared to *WT* brains (Fig 3A–D). Different types of progenitors, such as apical radial glia (aRG) or bRG cells, contribute to neuronal production in a direct or indirect way via the generation of IP cells (Florio & Huttner, 2014). To distinguish among NP subtypes, we stained mutant cortices for Pax6 (labeling aRGs next to the ventricular zone—VZ—or bRGs in subventricular zone—SVZ) and Tbr2 (a *bona fide* IP marker in SVZ). *Nr2f1 KO* cortices displayed a higher number of Pax6$^+$ RG cells from E12.5 to E17.5 compared to littermate controls (Fig 3E–G), suggesting enhanced

aRG self-renewal. Notably, IP production (a good indicator of the start of neurogenesis) was delayed, as mutant cortices showed a peak of Tbr2$^+$ cells at E16.5, instead of E15.5 as in *WT* brains (Fig 3H). Nevertheless, a high number of IPs were still maintained until birth (P0), indicating that IP-dependent neurogenesis is taking place at later time points in mutant brains compared to *WT* ones (Fig 3H). Interestingly, Pax6$^+$ Tbr2$^-$ bRGs, which are normally scarcely represented in murine brains (Reillo *et al*, 2011; Wang *et al*, 2011), doubled their numbers in mouse mutant cortices at late developmental stages (Fig 3F, F′ and I), possibly as a result of the overall Pax6$^+$ cell pool expansion. The identity of bRG cells was not only assessed by Sox2, Ki67, and Tbr2 triple labeling and by their basal position (Fig EV2A–G), but also by P-Vimentin, which labels the apical process of dividing bRGs (Fig EV2H and H′). A fraction of mitotic bRGs in the mutant basal SVZ, here referred to as oSVZ, also expressed low levels of Tbr2 (Fig EV2I–K), in accordance with previous studies and suggesting the presence of bRGs converting into IPs *in loco* (Hansen *et al*, 2010; Stahl *et al*, 2013). Interestingly, heterozygous (*HET*) embryos (lacking only one *Nr2f1* allele, hence more similar to BBSOAS patients) already showed a trend in increasing progenitor abundance during neocortical development (Fig EV3A–F). These results indicate that Nr2f1 regulates the number and local abundance of apical and basal NPs, notably Pax6$^+$ aRG and bRG cells, in a dosage-dependent manner.

We next wondered whether aberrant proliferation of mutant NP cells affects the shape and volume of the early mouse cortex. Early proliferating aRGs normally expand the neuroepithelium tangentially while leaving the VZ thickness unchanged (Caviness, 2003). Consistently, the early NP increase in mutant brains was reflected by cortical hemisphere expansion, as confirmed in whole brains and in brain sections (Fig 3J–K′). The hemisphere expansion was quantified by measuring the extension of the inner ventricle wall (dotted line in Fig 3K and K′). The hemisphere increased size was evident only in the posterior-most region of the cortex (average 78.9 ± 18.3% posterior increase; $P = 0.024$), consistent with Nr2f1 highest expression in posterior cortex (Fig 3L). Taken together, we found a regional increased number of NPs and an evident expansion of posterior hemispheres in early mouse *Nr2f1* mutant embryos.

The balance of symmetric versus asymmetric NP division is a fundamental process necessary to finely regulate brain neurogenesis and size (Götz & Huttner, 2005; Matsuzaki & Shitamukai, 2015). As the generation of Tbr2$^+$ basal IPs was delayed in mutants while the number of Pax6$^+$ cells was increased, we reasoned that *Nr2f1* loss could promote symmetric divisions, hence amplifying the apical progenitor pool at early time points. To confirm the self-renewing ability of mutant progenitors, we employed an *in utero*

electroporation (IUE) approach to directly visualize symmetrically dividing cells (self-renewing) and asymmetrically dividing ones (undergoing neurogenesis), by means of *Sox2-GFP* and *Tis21-RFP* reporter plasmid co-electroporations at E12.5 (Saade *et al*, 2013). NP cells which are dividing symmetrically to self-renew appeared as green cells (GFP$^+$), whereas neurogenic divisions producing two neurons were labeled in red (RFP$^+$) (Fig 3M). Asymmetrically dividing NP pairs, in which one daughter cell self-renews while the other one differentiates, expressed both plasmids resulting in a yellow signal (Fig 3M). *Nr2f1 KO* brains showed a higher proportion of self-renewing cells 18 h after IUE compared to *WT* electroporated littermates (Fig 3N–P), indicating an abnormally longer process of

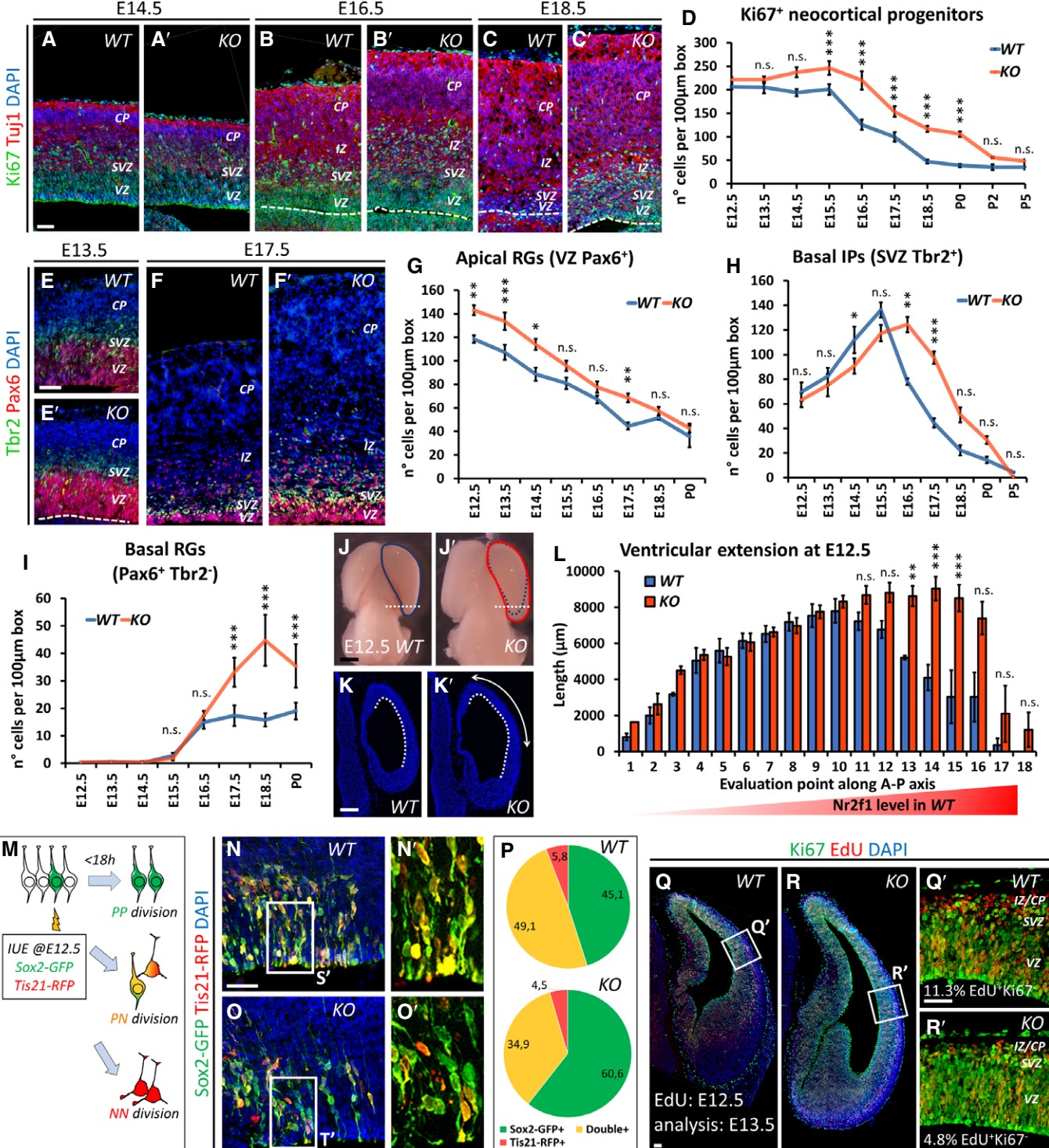

**Figure 3.**

**Figure 3. Nr2f1-mediated control of progenitor amplification *in vivo*.**

A–D  Ki67 (green; mitotically active progenitors) and Tuj1 (red; post-mitotic neurons) IF at E14.5 (A, A′), E16.5 (B, B′), and E18.5 (C, C′) of lateral pallia of wild-type (*WT*) versus mutant (*KO*) brains. Ki67$^+$ proliferating cells (quantified in D) accumulate in mutant cortices. n ≥ 3 brains.

E–E′  Tbr2 (green; IPs) and Pax6 (red; radial glia cells) IF of E13.5 (E, E′) and E17.5 (F, F′) lateral pallia of *WT* and *KO* brains. See Fig EV3 for Tbr2 Pax6 double staining in E17.5 *HET* embryos.

G–I  Quantification of different classes of cortical progenitors: apical radial glia cells (aRGs) as Pax6$^+$ NPs in VZ (G), basal intermediate progenitors (IPs) as Tbr2$^+$ cells in SVZ (H), and basal RGs as Pax6$^+$ cells in outer SVZ (I). See Fig EV3 for quantification of distinct progenitor classes in E17.5 *HET* embryos. n ≥ 3 brains.

J–L  E12.5 dorsal brain views (J, J′) and representative sections of posterior hemispheres (K, K′), showing a vesicle enlargement in mutant brains (K′). The extension of the ventricular surface was quantified along the A-P axis (L); red color code represents Nr2f1 gradient in *WT* brains. 12 μm-thick sections were collected on series of 10 slides; consecutive measurement levels are 120 μm apart from each other. n ≥ 3 brains.

M  Schematic representation of different aRG cell divisions, as visualized *in vivo* after co-*in utero* electroporation (IUE) of *Tis21-RFP* and *Sox2-GFP* plasmids into E12.5-old embryos. Green: symmetric proliferative divisions; yellow: asymmetric differentiative divisions; and red: symmetric differentiative divisions.

N–P  GFP (green; expressed under Sox2 promoter) and RFP (red; expressed under Tis21 promoter) IF of E13.5 lateral pallia, 18 h after IUE. The proportion of single- or double-positive NPs is shown in pie charts in (P). n = 3 electroporated brains.

Q–R′  Ki67 (green; progenitors) and EdU (red) IF of E13.5 *WT* (Q, Q′) and *KO* (R, R′) embryos, injected with EdU at E12.5. Differentiating cells (EdU$^+$Ki67$^-$) are located in the IZ/CP; percentages shown in (Q′, R′) were obtained from n ≥ 4 sections of n ≥ 2 brains. See Fig EV3 for neural differentiation index of E13.5 *HET* embryos.

Data information: Nuclei (blue) were stained with DAPI. In (D, G, H, I), the number of positive cells was quantified in 100 μm-width boxes, randomly placed across the LP. In graphs, data are represented as means ± SEM. Student's t-test (Q′, R′; E13.5, EdU injected at E12.5; *WT/KO*: *P = 0.03967) and two-way ANOVA (D, G, H, I, L, P) (*P < 0.05, **P < 0.01, ***P < 0.001). Scale bars: 50 μm. CP: cortical plate; IZ: intermediate zone; SVZ: subventricular zone; VZ: ventricular zone.

symmetric divisions upon *Nr2f1* loss. Moreover, in a short-term birth-dating analysis, cortices injected with EdU at E12.5 and then stained for Ki67 24 h later showed a lower proportion of EdU$^+$Ki67$^-$ cells exiting the cell cycle in *KO* embryos compared to *WT* littermates (Fig 3Q–R′). Consistent with previous data, *HET* embryos depicted an intermediate effect (Fig EV3G–J). These data strongly indicate that *Nr2f1* normally promotes asymmetric cell division and thus neurogenesis, and that in its absence NP cells divide symmetrically and expand their pool, ultimately delaying neurogenesis.

**Increased neuronal output in the posterior hemispheres of mouse *Nr2f1* mutant brains**

We then asked whether the highly proliferative NP pool observed in mouse mutant brains eventually undergoes neural differentiation, as suggested by delayed Tbr2 expression. To directly visualize differentiating cells, we injected EdU at given stages and evaluated the number of EdU$^+$Ki67$^-$ cells reaching the intermediate zone (IZ) or the cortical plate (CP) 24 h later (short-term EdU protocol; Fig EV4A–D). While remaining significantly longer in the cell cycle at early stages (E12.5; Figs 3Q–R′ and EV4A), mutant NPs generated a higher number of differentiating cells at later stages compared to *WT* (at E16.5 and P0; Fig EV4A–C′). However, when evaluating neural differentiation as a fraction of the total EdU$^+$ population, we found no difference between *WT* and *KO* after E15.5 (Fig EV4D). This suggests that the abnormal increased number of locally produced neurons after E15.5 (Fig EV4A) is an indirect consequence of an amplified NP pool, rather than of a higher tendency to undergo differentiation. Consistent results (high neuronal production at mid-late neurogenesis) were obtained with a long-term EdU injection approach, in which EdU was injected at different embryonic ages and then evaluated at P0 (Fig EV4E). To further reveal the type of neurons produced in *KO* cortices, we co-labeled EdU$^+$ cells with layer-specific markers and found that both early produced Tbr1$^+$ and later generated Cux1$^+$ neurons were over-represented in mutant brains (Fig EV4F–G″), whereas the number of Ctip2$^+$ neurons was not affected (Fig EV4F–F″). Notably, cells produced at E17.5 still included high number of neurons in the mutant brain, together with slightly increased

gliogenesis, as assessed by triple EdU GFAP NeuN labelling (Fig EV4H–J). As a consequence of intensive neurogenesis, mutant brains displayed a 27.1 ± 2.7% and 34.2 ± 1.7% increase in CP thickness at E17.5 and P0, respectively, and expanded occipital hemispheres (Fig EV4K–L″). Tuj1$^+$ post-mitotic neuron numbers confirmed an initial delay in neurogenesis followed by a significant neuronal increase in *KO* cortices at following stages (Fig EV4L, L′ and M), and leading to the formation of a thicker posterior CP with over-represented early- and late-generated (Tbr1$^+$ and Satb2$^+$) neurons (Fig EV4N–Q). Taken together, our data show that *Nr2f1* loss triggers a progenitor pool increase followed by a boost in neurogenesis, which correlates with an early lateral growth of the cortical neuroepithelium and a late radial expansion of the CP, globally leading to an occipital cortex enlargement.

**Region-specific control of cell cycle dynamics by Nr2f1**

As part of its role in regulating neurogenesis, we reasoned that Nr2f1 could control the cell cycle length itself by setting the average mitotic activity along the A-P neocortical axis. Key players of neocortical area mapping, such as Pax6 and Emx2, have been shown to finely regulate cell proliferation dynamics in a region-specific manner, linking areal identity to specific cell cycle durations, neurogenic activities, and cell fate acquisition (Dehay & Kennedy, 2007; Borrell & Calegari, 2014). By quantifying different classes of PH3$^+$ progenitors at a stage when both apical and basal progenitors are present, we found no difference in anterior/motor regions, while mutant brains showed increased mitoses in posterior/sensory neocortical regions, where Nr2f1 is normally highly expressed (Fig EV5A–C).

To directly measure the cell cycle duration time, we employed a double EdU/BrdU protocol in which pregnant females are injected with EdU first, followed by BrdU 1.5 h later, and then sacrificed at 2 h to process treated embryos (Martynoga *et al*, 2005; Mi *et al*, 2013) (Fig EV5D). The ratio of single EdU$^+$ or double EdU$^+$BrdU$^+$ cells over the total number of Ki67$^+$ cycling cells in the neocortical neuroepithelium (Fig EV5E–E″) was used to estimate the cell cycle time (Tc, the time needed to complete a full cell cycle) in E12.5 and E14.5 brains. While the cell cycle duration was not altered in

anterior-most regions, it was instead greatly reduced in more posterior *Nr2f1 KO* cortices (33.5 ± 1.7% shorter Tc), implying an accelerated cell cycle of E12.5 and E14.5 *KO* brains (Fig EV5F and F′), with *HET* brains already showing a trend toward cell cycle acceleration (Fig EV5G). As an alternative approach, we employed fluorescence-activated cell sorting (FACS), which evaluates the percentage of S-, G2- and M-phase cells after PH3 and propidium iodide staining (Appendix Fig S2). The number of cycling cells undergoing division (PH3$^+$) or entering S-and G2-phases was higher in E17.5 mutant posterior cortex compared to *WT* (Appendix Fig S2A–D‴), with little or no difference in anterior halves of cortices, in line with the *Nr2f1* graded expression profile. At E14.5, cell cycle progression was accelerated by 10% in *HET* posterior cortices and by 17% in *KO* ones, compared to control samples (Appendix Fig S2E–G). Overall, these results show that Nr2f1 dosage links positional identity of cortical progenitors to distinct cell cycle dynamics.

Based on these data, we decided to focus on the posterior/occipital cortex, where cellular abnormalities were more pronounced, for further cell cycle analysis. We wondered whether Nr2f1-dependent control of mitotic activity was taking place at specific time points. The total number of PH3$^+$ mitotic figures was higher in mutant cortices all along corticogenesis, especially from E14.5 to E17.5 (Fig 4A). By specifically quantifying mitoses in different compartments, we found increased divisions for both apical (Pax6$^+$PH3$^+$ in VZ) and basal (Pax6$^+$PH3$^+$ in oSVZ) RG cells. Tbr2$^+$ mitoses (SVZ) followed the same trend previously described for IP numbers, i.e., an initial delay in the mutant followed by high and sustained late activity (Fig 4B–D). To assess whether and how Nr2f1 controls cell cycle duration during corticogenesis, we performed double EdU/BrdU injection from E10.5 to E16.5 and found that the cell cycle of mutant progenitors was mainly accelerated during early/mid-neurogenesis (Fig 4E). An EdU cumulative labeling approach in E13.5 posterior cortices (Fig 4F–H), coupled with PH3 staining to visualize G2- and M-phase cells (Fig 4I–K), allowed us to precisely measure the duration of different cell cycle phases in VZ/SVZ NPs (Contestabile *et al*, 2009) (Fig 4L). Compared to *WT*, *Nr2f1 KO* NP cells had a slightly (14%) longer S-phase, typical of "younger" self-renewing progenitors (Calegari, 2005; Lange *et al*, 2009; Arai *et al*, 2011) (Fig 4L). S-phase lengthening was compensated by a significantly 31% shorter G1-phase in the mutant, which resulted in an overall shorter cell cycle duration (Fig 4L). Taken together, our data indicate that Nr2f1 operates a fine, region-specific control of cell cycle progression along the A-P axis, with strongest effects in the posterior/occipital neocortex. In its absence, cell cycle acceleration will contribute to altered cortical morphology, reminiscent of a posterior macrocephaly.

Finally, we investigated whether Nr2f1 levels influenced NP cell physiology also along the L-M neocortical axis, since Nr2f1 shows a lateral-high to medial-low gradient along the neocortical neuroepithelium, reaching highest levels in the latero-ventral region (Armentano *et al*, 2006; Faedo *et al*, 2008). While different Nr2f1 levels correlated with distinct degrees of neurogenesis and NP populations along the L-M axis (Appendix Fig S3A–E), this gradient was partially lost following Nr2f1 removal (Appendix Fig S3F). Interestingly, at E17.5, Sox2$^+$Pax6$^+$ bRG cells were extremely abundant in the DP (Appendix Fig S3G), which has been recently described as a gyrus-like region of the mouse cortex for its abundant presence of Hopx$^+$ bRG cell population (Vaid *et al*, 2018). On the contrary,

bRGs were scarcely represented in the medial (MP) and lateral (LP) pallia (Appendix Fig S3G). Upon Nr2f1 genetic inactivation, the L-M distribution of bRGs was altered (Appendix Fig S3H–K). Strikingly, the LP became populated by an abundant bRG cell population (Appendix Fig S3H, J and K), suggesting the acquisition of a more "dorsal-like" progenitor identity. Cell cycle length, as quantified by double EdU/BrdU injection at E14.5 (Appendix Fig S3L–O), was significantly accelerated in the LP of mutant brains (Appendix Fig S3O). The DP, characterized by intermediate Nr2f1 levels, showed a smaller but significant cell cycle acceleration (Appendix Fig S3O). Taken together, our data indicate that cell cycle progression and progenitor ratios are finely tuned by Nr2f1 dosage along both the A-P and L-M axes (Appendix Fig S3P).

## Nr2f1-mediated regulation of *Pax6* expression levels and cell cycle genes

Next, we aimed to identify some of the molecular mechanisms underpinning the high production of NP cells in posterior cortices of *Nr2f1 KO* mice. Pax6 plays a crucial role in controlling cell cycle and fate of both apical and basal NPs (Englund, 2005; Asami *et al*, 2011; Mi *et al*, 2013). Since persistent Pax6 expression in basal progenitors promotes proliferation and neurogenesis (Wong *et al*, 2015) and Pax6 levels can be modulated by Nr2f1 (Faedo *et al*, 2008; Tang *et al*, 2010), we hypothesized that abnormally increased Pax6 expression levels might contribute to the occipital expansion observed in *Nr2f1 KO* brains. Indeed, we found increased Pax6 protein in the LP of E12.5 and E14.5 mutant brains (Fig 5A–C; see also Fig 3E and E′), confirmed by a 33.3 ± 5.8% increase in *Pax6* transcript, as quantified by RT–PCR (Fig 5D). Interestingly, *Nr2f1 HET* brains already showed a significant increase in Pax6 expression compared to *WT* littermates (Appendix Fig S4A–E), even if not reaching the same level of *KO* ones. To assess whether Nr2f1 can modulate Pax6 expression in a cell-autonomous way, a *Nr2f1*-overexpressing plasmid was electroporated into E12.5 cortices and Pax6 signal intensity evaluated in GFP$^+$ cells 24 h later (Fig 5E–G″). GFP$^+$ progenitors with high Nr2f1 expression down-regulated Pax6 (Fig 5H), confirming that Nr2f1 cell-autonomously restrains Pax6 expression levels in cortical progenitor cells.

Furthermore, since *Nr2f1* mutant NP cells show accelerated cell cycle progression, we investigated the expression of several key cell cycle genes by RT–PCR on E12.5 posterior cortices (Fig 5I). While most genes failed to show any significant differences between E12.5 *WT* and *KO* brains, we nevertheless observed altered expression of some interesting candidates involved in cell proliferation and cell cycle control, such as *DCT*, *CycD1*, and *P21* (Fig 5I). The dopachrome tautomerase enzyme DCT has been previously demonstrated to be essential for promoting NP proliferation (Jiao *et al*, 2006); hence, its increase could be partially responsible for the increased mitotic index of mutant cells. Moreover, the expression of *cyclin D1* (*CycD1*), an essential regulator of the G1–S transition phase, was also affected in the mutant (43.7% lower compared to *WT*; Fig 5I). Since *CycD1* expression is particularly high during the G1-phase (Glickstein *et al*, 2007), its diminution could depend on the G1-phase shortening observed in *KO* cells (Fig 4L). Interestingly, *Nr2f1 KO* cortices revealed decreased *P21* expression compared to *WT* counterparts (Fig 5I). The cyclin-dependent kinase inhibitor P21 is considered an inducer of neurogenesis, as it is transiently

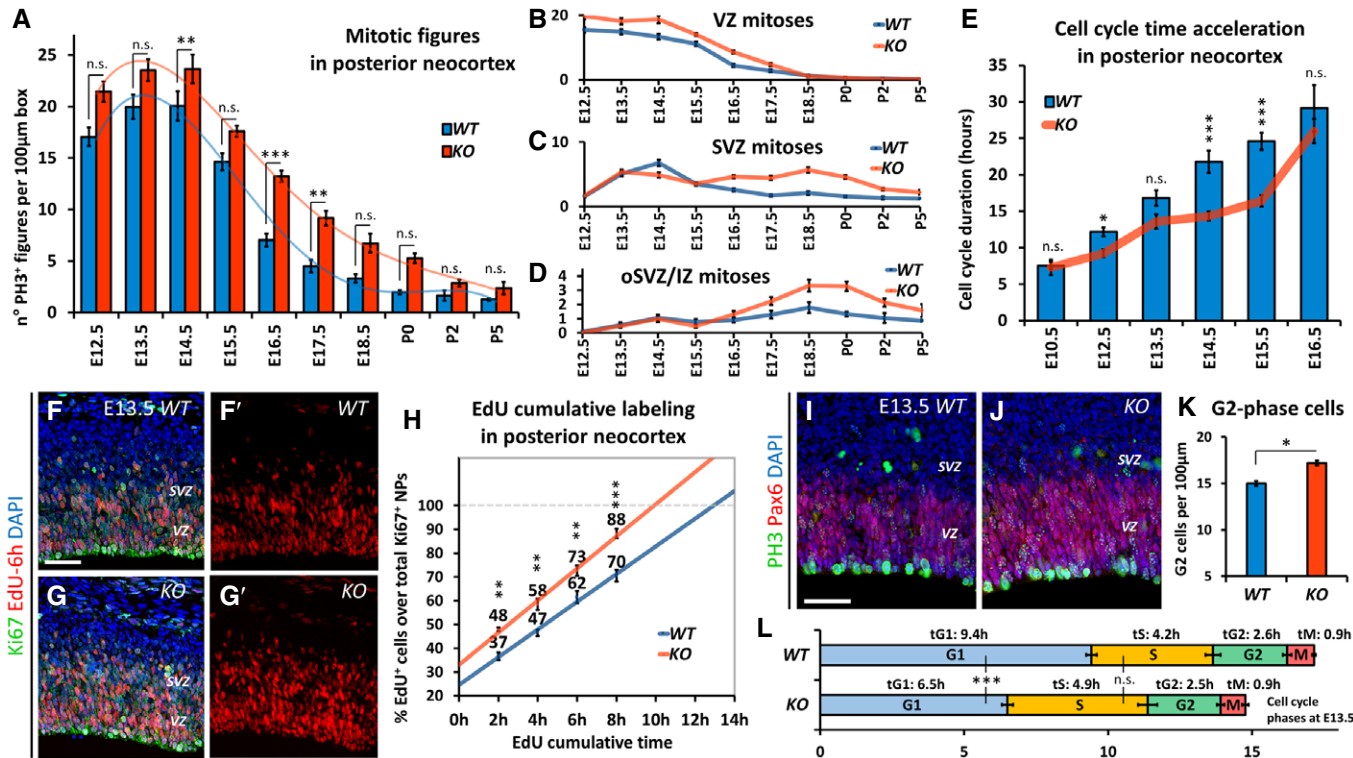

**Figure 4. Nr2f1 orchestrates cell cycle dynamics in posterior hemispheres.**

A–D	Graphs showing the average number of PH3⁺ mitotic figures in the posterior lateral pallium of *WT* and mutant cortices. The total number of mitoses (A) as well as the number of dividing cells in VZ (Pax6⁺Sox2⁺ aRGs; B), SVZ (Tbr2⁺ IPs; C), and outer (basal-most) region of SVZ (oSVZ/IZ; Pax6⁺ bRGs; D) is shown. For graphs (B-D), statistical analysis by two-way ANOVA is shown in Appendix Table S5. $n \geq 3$ brains per age/genotype.

E	Cell cycle duration at different embryonic ages (from E10.5 to E16.5) in *WT* (blue bars) or *KO* embryos (orange line), as quantified upon a double EdU/BrdU injection protocol in posterior-most LP (See Fig EV5D–E″ and Materials and Methods). $n \geq 3$ brains per age/genotype.

F–G′	Ki67 (green; NPs) and EdU (red) IF in *WT* (F,F′) and *KO* (G,G′) after 6 h consecutive EdU incorporation. Note the higher number of labeled NPs in *KO* cortices (G′) suggesting faster cell cycle progression and/or longer S-phase.

H	Best linear fit of EdU⁺Ki67⁺ NP percentage after 8 h EdU cumulative labeling in *WT* (blue line) and *KO* (orange line) brains. The x- and y-intercepts are proportional to the S-phase length and to the number of cycling cells, respectively, while the steepness of the line is proportional to cell cycle duration (see Materials and Methods; *WT*: y = 11.66x + 24.62; *KO*: y = 13.54x + 32.96). $n \geq 2$ brains per time point/genotype.

I–K	PH3 (green; dividing NPs) and Pax6 (red; aRGs) IF of E13.5 *WT* (I) and *KO* (J) lateral pallia. Strong and weak/punctate PH3 patterns show M-phase and G2-phase cells, respectively. G2-phase NP percentage is quantified in (K). As a result of shorter G1-phase and globally shorter cell cycle time, the percentage of G2-phase cells is increased in the *KO* NP pool. $n \geq 2$ brains.

L	Cell cycle time of E13.5 NPs in the lateral pallia of *WT* (upper column) and *KO* cells (lower column), as calculated by EdU cumulative labeling and PH3 staining (see Materials and Methods). *KO* NPs have a 16.1% faster cell cycle time, due to a 30.9% shorter G1-phase compensating a 14.3% longer S-phase. $n \geq 3$ brains.

Data information: Nuclei (blue) were stained with DAPI. In (A-E, H, K), the number of positive cells was quantified in 100 μm-width boxes, randomly placed across the lateral pallium. In graphs, data are represented as means ± SEM. Student's *t*-test (K; *$P < 0.05$) and two-way ANOVA (A-E, H, L) (*$P < 0.05$, **$P < 0.01$, ***$P < 0.001$). Scale bars: 50 μm. SVZ: subventricular zone; VZ: ventricular zone.

expressed in a subset of NPs to trigger cell cycle exit and differentiation (Siegenthaler, 2005). To support a causal relationship between Nr2f1 and P21, we acutely down-regulated *Nr2f1* via IUE of a CRISPR/Cas9 plasmid (herein called *PX458-αNr2f1*) in which the single guide RNA (*sgRNA*) recognizes the ATG starting codon of the *Nr2f1* gene, hence targeting and disrupting the sequence necessary for its translation (Appendix Fig S5). Nr2f1 expression was down-regulated or completely abolished in GFP⁺ electroporated cells (Appendix Fig S5A–C), indicating CRISPR/Cas9-mediated loss of one or two *Nr2f1* alleles, respectively. Upon *PX458-αNr2f1* IUE, we found a 59.5 ± 11.3% drop in the number of P21⁺ cells undergoing neurogenesis (Fig 5J–L). Together, we showed that Nr2f1 exerts its control on NP proliferation by modulating both cortical area

patterning genes (such as *Pax6*) and distinct cell cycle genes (such as *P21*) (Fig 5M).

Moreover, as *NR2F1* mutants display multiple deficits in addition to changes in cell cycle, we decided to employ a full transcriptomic profile analysis to have a more comprehensive picture of genetic changes upon *Nr2f1* loss. We compared *WT* and *KO* cortical transcriptomics by *RNA-Seq* at E15.5, a key stage of neurogenesis. Among differentially expressed genes (DEGs; see Appendix Fig S6 and Appendix Table S1), 102 and 68 were significantly up- and down-regulated, respectively (Fig 5N). Gene ontology (GO) analysis of DEG genes highlighted "neuron differentiation" and "neuron development" as most significant categories (Fig 5O; see Appendix Tables S2–S4 for complete results), in line with the role

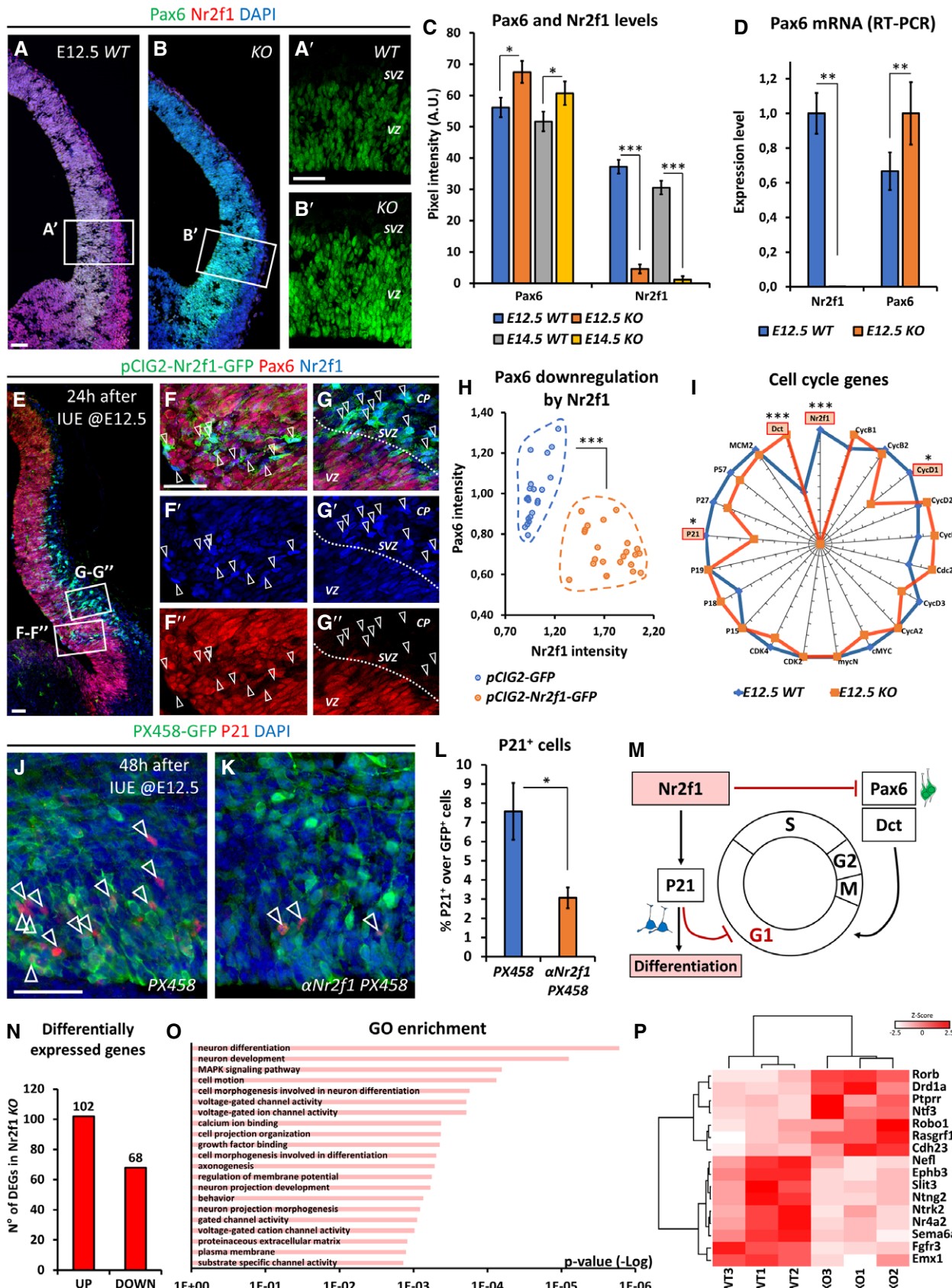

Figure 5.

**Figure 5.  Nr2f1-mediated molecular control of neural progenitors (NPs).**

A–B′   Pax6 (green; aRGs) and Nr2f1 (red) IF of E12.5 *WT* (A, A′) and *KO* (B, B′) lateral pallia. See Appendix Fig S4 for Pax6 and Nr2f1 staining and pixel intensity quantification in *HET* embryos.

C   Pax6 and Nr2f1 pixel intensity quantification at E12.5 and E14.5 indicating increased Pax6 levels upon Nr2f1 removal. $n \geq 3$ brains.

D   Real-time RT–PCR quantification of Nr2f1 and Pax6 expression in E12.5 cortices. $n \geq 3$ cortices.

E–G″   GFP (green; electroporated cells), Pax6 (red), and Nr2f1 (blue) IF of *pCIG2-Nr2f1-IRES-GFP* electroporated brains at E13.5 (24 h after IUE). VZ (F-F″) and CP (G-G″) regions are shown at high magnification. Arrowheads point to GFP$^+$ electroporated cells overexpressing Nr2f1 (blue in F′, G′), down-regulating Pax6 (F″), and rapidly migrating out of the VZ (G-G″).

H   Scatter plot showing Pax6 and Nr2f1 pixel intensity of VZ electroporated progenitors, comparing Nr2f1 overexpressing cells (orange dots) with control *pCIG2-GFP* electroporated cells (blue dots). Average Nr2f1 and Pax6 pixel intensities of the 2 populations were compared by two-way ANOVA and resulted significantly different (***$P$ = < 0.0001). $n$ = 2 electroporated brains.

I   Real-time RT–PCR of cell cycle genes comparing *WT* (blue line) and *KO* (orange line) cortices. *CyclinD1* and *P21* transcripts are down-regulated, whereas *Dct* is up-regulated in mutants. $n$ = 3 cortices.

J–L   GFP (green; electroporated cells) and P21 (red) IF of E14.5 cortex electroporated 48 h earlier with control *PX458* plasmid (J) or CRISPR/Cas9-expressing plasmid directed against Nr2f1 sequence (*PX458-αNr2f1*; K). Percentage of P21/GFP double-positive cells (L). Arrowheads in (J, K) point to P21$^+$ cells. $n$ = 2 electroporated brains.

M   Schematic model of Nr2f1 action on NP cell cycle and neural differentiation. Nr2f1 promotes neurogenesis by repressing Pax6 and Dct expression and thus cell cycle progression, and by activating P21-mediated cell cycle exit. Both actions modulate the G1-phase length.

N   Graph showing the number of differentially expressed genes (DEG; see Materials and Methods) up-regulated (UP) or down-regulated (DOWN) in *KO* compared to *WT*, as detected by *RNA-Seq* of E15.5 neocortices.

O   Gene ontology (GO) categories significantly enriched among DEG genes (*DAVID Gene Ontology software*; see Materials and Methods).

P   Hierarchical clustering and heatmap of the expression level of DEGs belonging to the "neuron differentiation" GO category in (O). Heat map color scale indicates normalized gene expression from high (red) to low (white) level.

Data information: In (A, B, J, K), nuclei (blue) were stained with DAPI. In (C), the pixel intensity was quantified in 100 μm-width boxes, randomly placed across the lateral pallium. Data are represented as means ± SEM. Student's *t*-test (L; *$P$ < 0.05) and two-way ANOVA (C, D, H, I) (*$P$ < 0.05, **$P$ < 0.01, ***$P$ < 0.001). For statistical analysis of *RNA-Seq* data (N-P), see Materials and Methods. Scale bars: 50 μm. SVZ: subventricular zone; VZ: ventricular zone.

played by Nr2f1 during neurogenesis. GO analysis by Panther Over-representation analysis or by Ingenuity pathway analysis showed similar results, additionally highlighting "axonal guidance signaling" and "netrin signaling" as highly regulated categories (Appendix Tables S3 and S4). The "neuron differentiation" GO category included known master genes of neurogenesis as well as cell adhesion, NP delamination, and axon guidance, such as *Rorb*, *Ephb3*, *Sema6a*, and *Slit3* (Fig 5P). Interestingly, the third GO category ("MAPK signaling pathway") suggests that Nr2f1 could orchestrate NP proliferation, fate, and differentiation in an indirect way by regulating signaling pathways with broader action (Begemann & Michon, 1995; Faedo *et al*, 2008), in addition to specific cell cycle or cortical master genes. Further functional studies are needed to validate Nr2f1 target genes and/or regulated pathways.

**Genetic manipulation of Pax6 levels rescues both *in vitro* self-renewal potential and *in vivo* cell cycle dynamics of Nr2f1-deficient NP cells**

To directly investigate whether Pax6 acts as a major effector downstream of Nr2f1 during cortical NP proliferation, we genetically decreased Pax6 expression levels by crossing a *Pax6 null* line (St-Onge *et al*, 1997) with the *Nr2f1* mutant line. To aim for a functional rescue, we looked for a decrease and not a full loss of Pax6 levels in a *Nr2f1 KO* background. First, we measured *Pax6 mRNA* levels in neurospheres obtained from E15.5 dissociated *Pax6* mutant cortices and confirmed progressive diminished levels in heterozygous (*Pax6 HET*) and *null* (*Pax6 KO*) neurospheres (Fig 6A). As expected, *Pax6* expression was increased in *Nr2f1 KO* neurospheres, and subsequent removal of one *Pax6* allele (*Nr2f1 KO, Pax6 HET*; Fig 6A) efficiently decreased *Pax6* transcripts in a *Nr2f1 KO* background. Interestingly, *Nr2f1 KO, Pax6 HET* neurospheres were smaller and proliferated for maximum up to 10 passages (Fig 6B

and C), indicating that Nr2f1-dependent modulation by Pax6 levels is key for NP self-renewal. To directly investigate the self-renewing ability of mutant progenitors, we dissociated neurospheres and plated NP cells at low density to perform a paired-cell test. Double Map2 and BLBP staining identified neuronal and progenitor cells, respectively, in newly dividing cell couples (Fig 6D–D″), assessing differentiation versus self-renewal. Isolated progenitors from *Nr2f1 KO* neurospheres generated a higher number of P-P (symmetric proliferative) cell divisions, at the expense of P-N (asymmetric) and N-N (symmetric differentiative) ones, confirming a higher tendency of mutant NPs to self-renew compared to *WT* ones (Fig 6E). However, when one *Pax6* allele was removed, cell division proportions went back to levels comparable to *WT* cells with a reduced number of symmetric proliferative divisions (Fig 6E). This indicates that Pax6 and Nr2f1 agonistically control the self-renewal efficiency of dividing NP cells in the developing neocortex.

We next wondered whether cell cycle progression was also affected in brains with high or low Pax6 levels. As for neurospheres, we removed one *Pax6* allele to restore almost normal protein levels in the LP of E12.5 *Nr2f1 KO* cortices (Fig 6F–H′; pixel intensity quantification in Fig 6I). By double EdU/BrdU incorporation, we evaluated the cell cycle duration in different mutants: while the cell cycle was shorter in *Nr2f1 KO* posterior cortices, it slowed back to a normal rate after removal of one *Pax6* gene copy (Fig 6J). Interestingly, *Pax6 HET* animals displayed an even longer cell cycle time compared to *WT* (Fig 6J), suggesting that, differently to what reported for more anterior cortical regions (Georgala *et al*, 2011; Mi *et al*, 2013), Pax6 can act as a cell cycle accelerator in posterior cortex. Finally, we tested whether Pax6 modulation could rescue Nr2f1-dependent control of NP cell division *in vivo*. We visualized the cleavage plane of dividing NPs by labeling them for P-Vim and γ-Tubulin to evaluate their division angle (Fig 6K). A vertical cleavage plane has been associated with symmetric divisions originating two

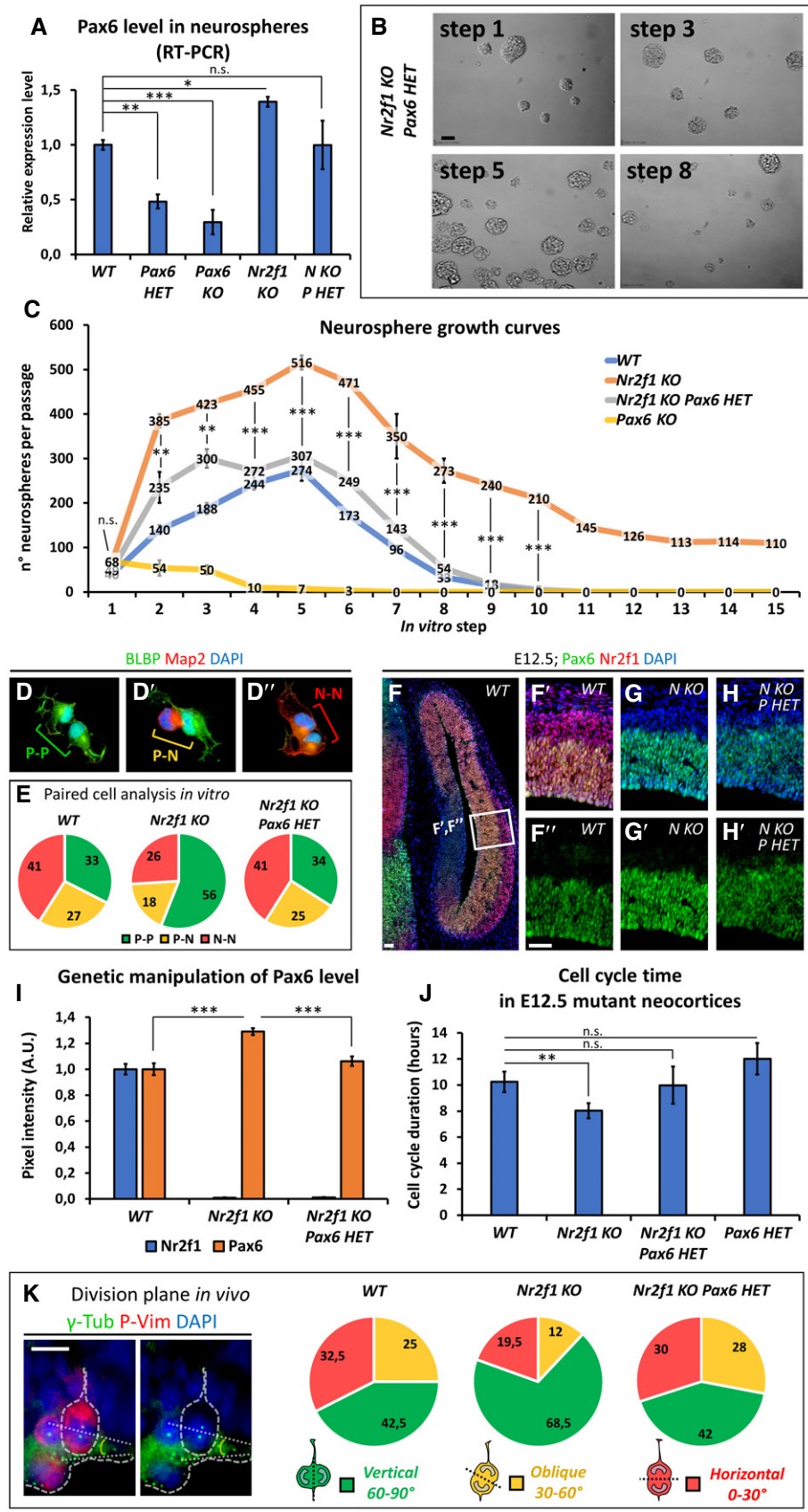

Figure 6.

**Figure 6. Functional rescue of NP cell cycle dynamics via Pax6 genetic modulation.**

A   Real-time RT–PCR analysis of *Pax6* expression in neurospheres with different genotypes, as indicated. *n* = 3 cortices. N KO, P HET: *Nr2f1 KO, Pax6 HET*.

B   Representative images of *Nr2f1 KO Pax6 HET* neurospheres obtained from E15.5 neocortices and cultured *in vitro*.

C   Graph showing the number of neurospheres per P100 cell plate, at different steps, as indicated. While *Nr2f1 KO* neurospheres (orange line) proliferate for a longer time compared to *WT* ones (blue line), the loss of one *Pax6* allele (gray line) almost restores normal proliferation rate and exhaustion time. Complete *Pax6* loss is not compatible with stem cell renewal (yellow line). *n* ≥ 3 culture wells from *n* = 2 independent batches.

D, E   Paired-cell analysis after neurosphere dissociation and 24-h culture at clonal density. Couples of dividing cells were identified by Map2 (red; N, neurons) and BLBP (green; P, progenitors) IF. Pie charts in (E) show the proportion of P-P (proliferative; D), P-N (asymmetric differentiative; D′), or N-N (symmetric differentiative; D″) couples in *WT*, *Nr2f1KO*, and *Nr2f1KO Pax6 HET* animals. *n* ≥ 3 samples from *n* = 2 culture batches.

F–I   Pax6 (green) and Nr2f1 (red) IF in the posterior-most region of E12.5 cortex, showing Pax6 upregulation in *Nr2f1 KO* animals (G, G′) compared to *WT* (F′, F″). Normal Pax6 levels can be restored by loss of one Pax6 allele (*Nr2f1 KO, Pax6 HET*; H, H′). Pixel intensity quantification of Nr2f1 (blue) and Pax6 (orange) is shown in (I). *n* ≥ 3 brains. N KO: *Nr2f1 KO*; P HET: *Pax6 HET*.

J   Cell cycle duration at E12.5 in different mutants as indicated, quantified in posterior regions with double EdU/BrdU injection protocol (see Materials and Methods). *n* ≥ 6 sections from *n* = 2 brains.

K   Insets showing representative γ-Tubulin (green) and phospho-Vimentin (red) IF of the cortical VZ surface to evaluate the orientation of the cleavage plane of dividing aRGs. Pie charts show the percentage of vertical (green), oblique (yellow), and horizontal (red) division planes of mitotic figures in the E12.5 lateral pallium of animals with different genotypes, as indicated. *n* ≥ 2 brains per genotype.

Data information: Nuclei (blue) were stained with DAPI. In (I, J), the pixel intensity or the number of positive cells was quantified in 100 μm-width boxes, randomly placed across the LP. Data are represented as means ± SEM. Two-way ANOVA (A, C, E, I, J, K; *$P < 0.05$, **$P < 0.01$, ***$P < 0.001$). For statistical analysis of (E, K), see Appendix Table S5. Scale bars: 50 μm.

identical pluripotent daughter cells, whereas oblique or horizontal cleavage planes are mainly linked to neurogenesis, as they give rise to basal progenitors and neurons (Noctor *et al*, 2008; LaMonica *et al*, 2013; Florio & Huttner, 2014; Matsuzaki & Shitamukai, 2015). *Nr2f1 KO* NPs tended to divide more often through a vertical cleavage plane compared to *WT* ones, indicative of increased self-renewal activity (Fig 6K). Upon loss of one *Pax6* allele, the division plane of dividing NPs *in vivo* was however restored to levels similar to *WT* (Fig 6K). Together, our data indicate that key proliferative features of *Nr2f1* mutant progenitors can be rescued by lowering Pax6 levels. This implies that a complex network of interconnected key area mapping genes is required for the correct neurogenic activity to take place in a time- and region-specific manner.

### NR2F1 controls neurogenesis in human brain organoids

To investigate whether human NR2F1 is also able to regulate neurogenesis, we used *in vitro* 3D cerebral organoids, which recapitulate some of the major characteristics of early human brain development (Lancaster *et al*, 2013; Wang, 2018; Klaus *et al*, 2019). Immunostaining of human iPS-derived organoids showed increased NR2F1 expression levels from day 30/40 to day 70 of *in vitro* differentiation (Fig 7A–C), as well as distinct protein levels with higher expression in NPs than in post-mitotic neural cells (Fig 7B–B″), in accordance with our previous expression data (Fig EV1G′, G″, I′ and I″). Notably, day 40 neuroepithelia also displayed distinct NR2F1 protein levels in function of PAX6 expression (low NR2F1 level in high PAX6-expressing neuroepithelia, and *vice versa*; Fig 7D–G), suggesting that the complementary expression profile between NR2F1 and PAX6 is most likely an evolutionary conserved process.

To start investigating NR2F1 role in human progenitors, we electroporated the CRISPR/Cas9 plasmid (*PX458-αNR2F1*) into day 33-cultured cerebral organoids, taking advantage of the high (100%) sequence conservation between mouse and human, and analyzed NR2F1 expression 7 days later (Fig 7H–J). The control experiment consisted in electroporating the empty *PX458* plasmid, without *sgRNA* sequence directed against *NR2F1* sequence. As in mouse cortex, the plasmid efficiently down-regulated NR2F1 expression

(Fig 7I and J), confirming that the *PX458-αNR2F1* vector can modulate NR2F1 expression both in mouse brain and human organoids. Next, we evaluated protein levels of the NP marker SOX2 and of the neuronal marker TUJ1 in GFP⁺ control- and *PX458-αNR2F1*-electroporated cells. We found that decreased NR2F1 levels were associated with increased tendency of progenitors to remain in the cell cycle and express SOX2 or PAX6, together with a reduced differentiation rate of double TUJ1⁺GFP⁺ cells (Fig 7K–N). Indeed, the number of GFP⁺ progenitor cells was increased at the expense of TUJ1⁺ 7 days after IUE (Fig 7M), similarly to what previously shown in mouse embryos. Overall, this is a first step demonstrating the evolutionary conservation between mice and humans of NR2F1 as a major regulator of cortical neurogenesis through modulation of the NP pool and PAX6 expression levels.

## Discussion

Altered balance of cell proliferation, differentiation and neuron migration can lead to cortical malformations, often associated with a huge variety of clinical manifestations, such as intellectual disability (ID), schizophrenia, epilepsy, and/or autism spectrum disorder (Barkovich *et al*, 2012; Sun & Hevner, 2014; Parrini *et al*, 2016; Juric-Sekhar & Hevner, 2019). Thanks to human genetic and animal model studies, it is now well accepted that macrocephaly, polymicrogyria, and/or microcephaly can be due to alterations in progenitor cell behavior (Boland *et al*, 2007; Finding of Rare Disease Genes (FORGE) Canada Consortium *et al*, 2012; Lee *et al*, 2012; Poduri *et al*, 2012; Mirzaa *et al*, 2013; Marchese *et al*, 2014; Terrone *et al*, 2016), as also recently confirmed by 3D-cell culture assays (Lancaster *et al*, 2013; Li *et al*, 2017). Moreover, advances in DNA sequencing technologies and concurrent improved resolution of neuroimaging techniques have allowed the identification of more genes and signaling pathways underlying different types of MCDs (Juric-Sekhar & Hevner, 2019). In this study, we aimed to investigate the cellular and molecular mechanisms underlying brain malformations of an emerging neurodevelopmental disease, called BBSOA syndrome, in which patients (mainly children) show clinical

features of ID, epilepsy, repetitive behaviors, speech problems, and autistic-like traits. A previous report focused on the optic atrophy defect (Bertacchi *et al*, 2019), an important component of the clinical phenotype (Bosch *et al*, 2014; Chen *et al*, 2016), but nothing was known on whether *NR2F1* haploinsufficiency would lead to particular types of cortical malformations. In this study, we describe

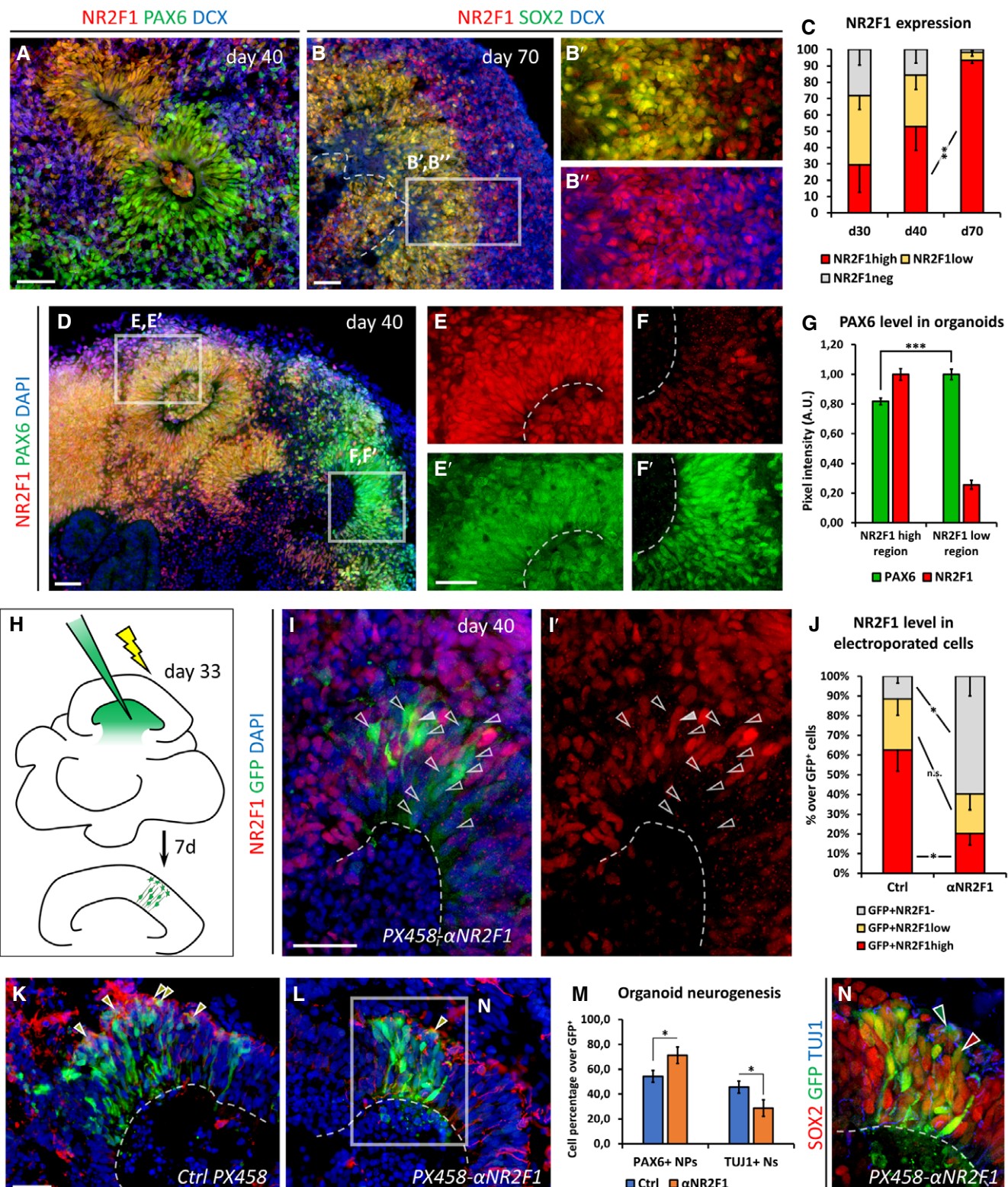

Figure 7.

**Figure 7.  Delayed neurogenesis upon NR2F1 down-regulation in human brain organoids.**

A–C    NR2F1 expression in cerebral organoids during neural induction and differentiation *in vitro* (day 40 and day 70, respectively), together with the NP markers PAX6 (green in A), SOX2 (green in B, B'), and DCX as a neural marker (blue in A, B, B''). Quantification of NR2F1 levels in NPs is shown in (C). $n \geq 4$ organoids from $n = 2$ batches.

D–G    NR2F1 (red) and PAX6 (green) IF in day 40 organoids. Neuroepithelia with high NR2F1 levels show weak PAX6 expression (E, E'), and *vice versa* (F, F'). Quantification by pixel intensity analysis (G). $n \geq 4$ organoids from $n = 2$ batches.

H      Schematic representation of an organoid electroporation. *PX458* plasmids were injected in ventricular-like cavities, and organoids were electroporated and processed after 7 days of *in vitro* culture.

I, J    NR2F1 (red) and GFP (green) IF in day 40 human brain organoids upon electroporation of the *PX458-αNR2F1* plasmid. In (I, I'), GFP$^+$NR2F1$^-$ cells (empty arrowheads) and a GFP$^+$NR2F1$^+$ cell (white arrowhead) are shown; quantification in (J). $n \geq 6$ organoids from $n = 2$ batches.

K–M    TUJ1 (red; differentiating neurons) and GFP (green) IF 7 days after electroporation of control (K) and *PX458-αNr2f1* plasmids (L). Co-expression of GFP with SOX2 or TUJ1 distinguishes neural progenitors (NPs) from neurons (Ns), as quantified in (M). Arrowheads in (K, L) point to TUJ1$^+$ GFP$^+$ cells. $n \geq 6$ organoids from $n = 2$ batches.

N      Red and green arrowheads in (N) point to a SOX2$^+$ (red) progenitor and a TUJ1$^+$ (blue) neuron, respectively.

Data information: Nuclei (blue in D, I, K, L) stained with DAPI. In (C, G, J), the pixel intensity or the number of positive cells was quantified in 100 μm-width boxes, randomly placed across the cortex/organoid neuroepithelia, while in (M) the number of positive cells was normalized over the total number of GFP$^+$ cells. In graphs, data are represented as means $\pm$ SEM. Two-way ANOVA test (*$P < 0.05$, **$P < 0.01$, ***$P < 0.001$). Scale bars: 50 μm.

a new cohort of BBSOAS patients who show a reproducible area-specific folding defect resembling polymicrogyria and restricted to supramarginal and angular gyri, a region known to be involved in a number of processes related to language, spatial cognition, memory retrieval, attention, and number processing (Stoeckel *et al*, 2009; Oberhuber *et al*, 2016). NR2F1 is expressed in a modular fashion along primary convolutions, particularly in neurogenic bRG cells, and its down-regulation affects neural differentiation in human brain organoids. In the mouse, precise levels of Nr2f1 are necessary for maintaining the delicate balance between proliferative and differentiative cells via the control of key patterning genes and cell cycle modulators. Thus, we propose NR2F1 as a novel regulator of cell proliferation and human cortical folding acting predominantly in an area-specific manner during early development.

**Regional modulation of the progenitor pool size by a Nr2f1 expression gradient**

Our data show that precise Nr2f1/NR2F1 levels along the A-P axis correlate with specific profiles of progenitor abundance and their neurogenic activity in both mouse and human brains. In this study, we thoroughly assessed the effects of mouse Nr2f1 removal on cell cycle progression in the posterior cortex, where Nr2f1 is expressed at its highest level (Armentano *et al*, 2006; Tomassy *et al*, 2010). Nr2f1 promotes asymmetric divisions by regulating cell cycle progression, hence modulating the progenitor pool amplification during cortical development. Upon loss of Nr2f1, the posterior-most region undergoes extensive and prolonged proliferation, followed by delayed neurogenesis and high neuronal production, ultimately leading to extended occipital hemispheres (model in Fig 8A and B). The expansion of occipital convolutions described in P1 and P2 is

somehow reminiscent to the posterior macrocephaly we describe in the mouse and could be the result of similar mechanisms of progenitor pool expansion, due to cell cycle acceleration and delayed neurogenesis. Thus, we propose that precise expression levels of NR2F1 are fundamental for the correct and area-specific balance between progenitor self-amplification and neurogenesis. Indeed, intermediate Nr2f1 expression levels in the dorsal pallial region, recently described as a gyrus-like structure and characterized by an abundance of Hopx$^+$ cells (Vaid *et al*, 2018), seem to be appropriate to trigger efficient neurogenesis, while concomitantly allowing basal progenitor amplification (summarized in Appendix Fig S3P). In human embryos, NR2F1 expression initially spans the A-P and L-M axes of the developing neocortex, but local modules of precise NR2F1 levels are associated with distinct degrees of neurogenesis and bRG populations during formation of primary convolutions. Hence, medium dosage of NR2F1 localized in developing folds could provide a "neurogenic boost", which would trigger high neuronal output while allowing sustained cell proliferation, both necessary for cortical expansion and folding (Fig EV1M). We thus propose NR2F1 as a novel modulator of progenitor cell production, including bRGs, in distinct localized regions.

However, our finding that *Nr2f1* loss in the mouse leads to an amplification of progenitors, notably aRG and bRG cells, seems apparently contradictory to the initial observation that NR2F1 in human cortex is highly expressed in bRG-rich gyri. The two observations can be reconciled by interpreting the role of NR2F1/Nr2f1 as a pro-neurogenic factor. Medium–high levels of human NR2F1 in cortical folds correlate with high abundance of bRGs and thus with high neurogenesis, whereas low NR2F1 levels in sulci are associated with low neurogenesis. In the mouse, complete Nr2f1 loss results in prolonged proliferation and delayed neural differentiation,

**Figure 8.  Schematic representation of cortical cellular and morphological consequences upon Nr2f1 loss.**

A    Nr2f1 is expressed along an anteromedial-low to posterolateral-high gradient (red color code) in the developing mouse cortex, spanning from VZ aRGs to SVZ IPs, bRGs, and CP neurons. Upon Nr2f1 removal (*KO*), the NP pool of the posterior cortex expands leading to occipital macrocephaly.

B    Early to late effects on progenitor and neuron behavior upon Nr2f1 deficiency. Early in development (E10.5-E12-5), Nr2f1 loss causes cell cycle acceleration, increased NP self-renewal, and sustained Pax6 expression. Neurogenesis (as well as the expression of neurogenetic factors such as Tbr2 and P21) is delayed, thus allowing amplification of the progenitor pool and lateral expansion of the posterior hemispheres. At later stages (E13.5-E14.5), neurogenesis comes into play and neurons are produced at high rate. Persistent Pax6 expression in the expanded NP population leads to abundant basal RG production at late time points (E16.5-E18.5). High neuronal output results in radial expansion and generation of a thick posterior cortex.

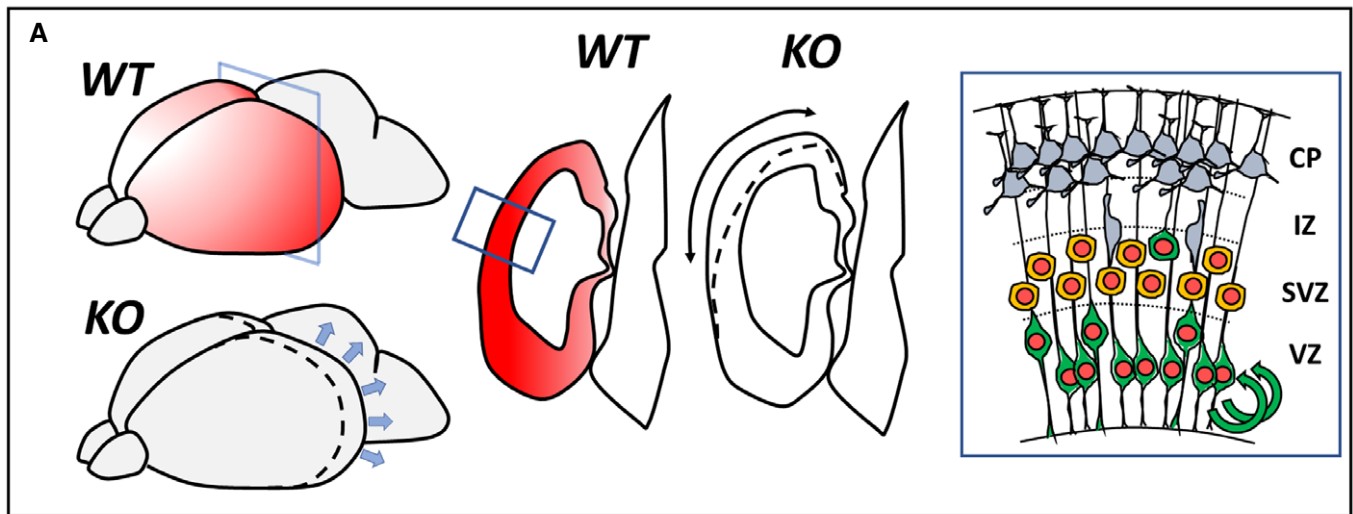

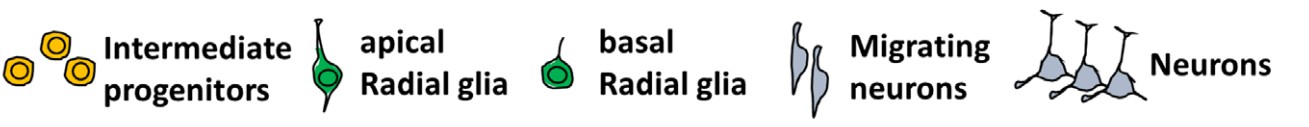

**Figure 8.**

consistent with the idea that Nr2f1 is promoting neurogenesis. Hence, the specific consequence of murine *Nr2f1* loss on radial glia amplification represents an indirect effect of enhanced cell proliferation associated with sustained Pax6 levels and is thus only apparently controversial with high NR2F1 expression in human bRG cells.

### The role of Nr2f1 in cell cycle control

Previous reports have linked cell cycle progression to fate acquisition. Neural progenitors with longer S-phase and shorter G1-phase undergo self-renewal instead of neurogenesis (Calegari, 2005; Lange *et al*, 2009; Arai *et al*, 2011). Consistently, neural progenitors in *Nr2f1* mutants, showing accelerated cell cycle due to a shorter G1-phase, increase the progenitor pool by symmetric division, while neurogenesis is delayed. Cyclin-dependent kinase inhibitor P21 is normally up-regulated in NPs lengthening the G1-phase and exiting the cell cycle to differentiate (Siegenthaler, 2005; Buttitta & Edgar, 2007; Heldring *et al*, 2012); hence, P21 down-regulation in Nr2f1-deficient progenitors could explain a delay in neurogenesis observed in mutant brains. Interestingly, we found that specific cell cycle durations are associated with regional abundance of progenitor subtypes, such as bRG cells in the dorsal pallium. Consistently, a link between increased proliferation and expansion of basal progenitor population, ultimately resulting in macrocephaly, was demonstrated in a mouse model in which the cell cycle progression was artificially and transiently accelerated (Nonaka-Kinoshita *et al*, 2013). It is tempting to speculate that specific cell cycle dynamics might be required for regionalized bRG amplification and that Nr2f1 levels could be key for controlling both cell cycle time and apico-basal progenitor ratio in an area-specific fashion.

### NR2F1-mediated control of PAX6 in mouse and human brains

Our molecular characterization indicates that Nr2f1/NR2F1 inhibits Pax6/PAX6 in mouse and human brain primordia, respectively. In mouse, Pax6 is known to act not only as an area patterning gene but also as a key regulator of cell cycle duration (Mi *et al*, 2013; Estivill-Torrus *et al*, 2002), cell division plane (Asami *et al*, 2011), and progenitor activity (Englund, 2005; Wong *et al*, 2015). Sustained Pax6 expression generates bRG in the developing neocortex (Wong *et al*, 2015), in line with our data. However, it must be noted that Pax6 seems to act as a break for cell cycle progression (Georgala *et al*, 2011), which is then accelerated in *Pax6 Sey* mice (Mi *et al*, 2013). On the contrary, we found that lowering Pax6 expression slows down cell cycle progression and restores normal cell cycle time in the posterior cortex of *Nr2f1* mutants. Hence, Nr2f1 and Pax6 might play different roles in function of their reciprocal area-specific co-expression levels, as previously suggested for Pax6 (Bishop, 2000; Bishop *et al*, 2002; Zembrzycki *et al*, 2007; Sansom *et al*, 2009; Asami *et al*, 2011). High Pax6 expression in anterior cortex directly correlates with low Nr2f1 expression, and *vice versa* in posterior cortex, and a complementary expression pattern between NR2F1 and PAX6 can also be reproduced in human cerebral organoids. Accordingly, functional modulation of Nr2f1 affects Pax6 expression in mouse, in line with previous findings (Faedo *et al*, 2008; Tang *et al*, 2010).

Finally, while endogenous Nr2f1 levels follow the neurogenic gradient along the L-M axis, this is not the same for the A-P axis;

the occipital-most cortex, expressing highest *Nr2f1* levels, has a delayed development compared to the anterior cortex with very low levels of this gene. We believe that *Nr2f1* gradient expression along the A-P axis is more related to acquiring a distinct areal identity than to modulating neurogenesis, in line with its recognized role as a "sensory" mapping gene (Alfano *et al*, 2014). *Pax6* would be instead responsible for promoting neurogenesis in the anterior-most cortex by controlling cell cycle parameters, as previously shown (Mi *et al*, 2013). It is well recognized that the regulation of the tangential extent of individual areas, as well as their precise neuronal pool, is under the control of cell cycle duration and rate of cell cycle re-entry of progenitors (Lukaszewicz *et al*, 2005). Hence, reciprocal regulation of early cortical master genes, such as *Pax6* and *Nr2f1*, would control the regional expansion of progenitor cell types, ultimately orchestrating regional and area-specific growth (Cadwell *et al*, 2019).

### Regional gyral pattern abnormalities in NR2F1-haploinsufficient patients

In this study, we report six novel cases of patients carrying NR2F1 mutations who are affected by ID and other cognitive disorders (Table 1). Patients display multiple cognitive symptoms together with optic nerve atrophy, a key feature of BBSOAS (Bosch *et al*, 2014; Chen *et al*, 2016). Here, we expand the clinical features of the BBSOA syndrome by describing for the first time some subtle but consistent brain malformations, such as aberrant cortical folding of the supramarginal and angular gyri and elongated occipital convolutions, in a new cohort of BBSOAS patients. Some phenotypic traits described in the six new patients, such as speech difficulties, stereotypical movements, and hyperactivity, are consistent with malformations in those specific regions (Stoeckel *et al*, 2009; Oberhuber *et al*, 2016). In spite of the impossibility to obtain high-resolution MRI images for all patients, we speculate that similar brain malformations might be shared among other BBSOAS patients. The abnormal gyral pattern we have observed in our cohort of patients, which is not associated with cortical thickness abnormalities, does not completely fall under the classical description of PMG or other MCDs (Manzini & Walsh, 2011; Barkovich *et al*, 2012). However, canonical MCD classifications proposed to date not always encompass the high morphological heterogeneity described in patients with abnormal microgyria (Guerrini & Dobyns, 2014).

A direct implication of NR2F1 in the development and positioning of cortical folds could not be tested in the mouse model. In fact, lissencephalic brains, such as the murine one, are not suitable and/or permissive systems to interrogate cortical folding, as they normally develop a smooth surface. While manipulation of few genes can nevertheless trigger cortical convolutions (Stahl *et al*, 2013; Florio *et al*, 2015; del Toro *et al*, 2017) and/or greatly amplify basal progenitor abundance (Borrell, 2018), genetic misexpression more often results in a general increase or decrease of the murine cortical size, without surface bending (Groszer, 2001; Kwon *et al*, 2006; Lange *et al*, 2009; Tokuda *et al*, 2011; preprint: Gompers *et al*, 2016). Consistently, down-regulating *Nr2f1* in the mouse leads to occipital macrocephaly by acting on cell cycle progression and progenitor pool size, but these changes were not selectively affecting the bRG population and thus inducing cortical

gyrification. Genes inducing folding in gyrencephalic species have been shown to be expressed in local modules, predictive of gyri and sulci positioning (de Juan Romero *et al*, 2015). Accordingly, human NR2F1 is both expressed along macro-gradients (A-P and L-M axes) and locally modulated in micro-modules; hence, NR2F1 could provide a link between positional identity of distinct neocortical areas and local formation of gyri and sulci. We speculate that abnormally low levels of *NR2F1* could lead to accelerated cell cycle progression and altered ratio of distinct progenitor subtypes, resulting in specific types of cortical malformations, such as elongated occipital convolutions or PMG-like abnormalities. However, to directly demonstrate NR2F1 involvement in cortical folding, more experimental approaches will be required in the future, such as the use of ferret, a gyrencephalic animal model previously employed for challenging the formation of cortical convolutions (Nonaka-Kinoshita *et al*, 2013; de Juan Romero *et al*, 2015; Fernández *et al*, 2016).

## Conclusion

Our BBSOAS mouse model has disclosed increased proliferative progenitor activity in the occipital cortex, due to sustained Pax6 expression together with delayed P21-dependent block of cell cycle progression. Even if the involvement of a similar genetic pathway needs to be validated in human brains, our first data on brain organoids (this study, and M. Bertacchi and M. Studer, manuscript in preparation) point to that direction.

# Materials and Methods

### Study approval

The individuals and families with BBSOAS were recruited through multiple French clinical genetics centers. Written informed consent was obtained from the legal guardians of the probands, according to the Declaration of Helsinki. All experiments involving the use of human samples conformed to the principles set out in the WMA Declaration of Helsinki and the Department of Health and Human Services Belmont Report. Fetal tissues were kindly provided by Cécile Allet and Paolo Giacobini (Lille, France) and were made available in accordance with French bylaws (Good practice concerning the conservation, transformation, and transportation of human tissue to be used therapeutically, published on December 29, 1998); studies on human fetal tissue were approved by the French agency for biomedical research (See Bertacchi *et al*, 2019; "Agence de la Biomédecine, Saint-Denis la Plaine", France, protocol no.: PFS16–002). All mouse experiments were conducted in accordance with relevant national and international guidelines and regulations (European Union rules; 2010/63/UE) and have been approved by the local ethical committee in France (CIEPAL NCE/2019-548).

### Mouse cell culture

Embryonic neocortical tissue was isolated at E12.5 or E15.5, dissociated, and cultured in neurospheres following previous protocols (Reynolds *et al*, 1993; Brewer & Torricelli, 2007). As an alternative to neurosphere culture, NPs were dissociated by trypsin digestion

and seeded as a monolayer on a Matrigel-coated plastic surface (Conti *et al*, 2005).

### Human brain organoids

Human induced pluripotent stem (iPS) cells and brain organoids were obtained and cultured as previously described (Lancaster *et al*, 2013; Klaus *et al*, 2019).

### Immunofluorescence

Mouse embryonic brains/whole heads were processed for immunostaining, as previously described (Armentano *et al*, 2006, 2007; Terrigno *et al*, 2018). Primary and secondary antibodies used are listed in detailed Materials and Methods (see Appendix File). Images were acquired at an Apotome Zeiss, using the AxioVision software.

### EdU/BrdU injection and cell cycle time calculation

The nucleoside analogs EdU (5-ethynyl-2′-deoxyuridine) and BrdU (5-bromo-2′-deoxyuridine) were both injected at 10 mg/kg intraperitoneally in pregnant females carrying embryos at the desired age. For the calculation of cell cycle time, we followed a previously described protocol (Martynoga *et al*, 2005). For cumulative EdU cell cycle quantification (Nowakowski *et al*, 1989; Takahashi *et al*, 1993; Contestabile *et al*, 2009), E13.5 embryos were labeled with multiple EdU injections (at 0, 2, 4, and 6 h). In all protocols listed, EdU was detected using EdU click-iT technology (Invitrogen, C10340), while BrdU required acid antigen retrieval (30 min at 37°C in 2N HCl with 0.5% Triton X-100) followed by immunostaining (Sigma B8434, 1:1,000, mouse). Detailed version of these protocols can be found in the Appendix File.

### *In utero* electroporation (IUE)

IUE was performed on E12.5 mouse brains targeting the latero-dorsal regions of the telencephalic dorsal pallium. Conditions of electroporation have been previously described (Parisot *et al*, 2017; Terrigno *et al*, 2018). The *Sox2p-GFP* and *Tis21p-RFP* plasmids were a kind gift of E. Marti and G. Le Dréau (Saade *et al*, 2013). For *Nr2f1* knock-out/knock-down, we used a CRISPR/Cas9 vector (Ran *et al*, 2013) in *wild-type* background mice. The anti-Nr2f1 *sgRNA* probe for the CRISPR/Cas9 construct (cloning primers: CACCgcga gatccgcaggacgacg; AAACcgtcgtcctgcggatctcgc) was selected and checked for specificity *in silico* with Crispor (http://crispor.tefor.ne t/) and then cloned in the PX458 empty vector (Addgene; pSpCas9 (BB)-2A-GFP).

### Electroporation of organoids

Human brain organoids were kept in neural differentiation medium without antibiotics for 2 h before the experiment. The organoids were then placed in an electroporation chamber (Harvard Apparatus, Holliston, MA, USA), and the plasmid DNA was injected at a concentration of 1 μg/μl in several positions (Cárdenas *et al*, 2018; Klaus *et al*, 2019). The organoids were then subjected to five pulses at 80 V with a 50-ms duration in an interval of 500 ms using an ECM830 electroporation device (Harvard Apparatus).

### Cell cytometry

Neocortices were dissected on ice and digested using a Worthington Papain Dissociation System (Serlabo technologies LK003150), following manufacturers' instructions. Low-bind Eppendorf tubes were used to limit material loss. Dissociated single cells were washed twice with ice-cold PBS 1X, then fixed by slowly adding ethanol while mixing on a vortex, and stored at $-20°C$. For cytometric analysis, cells were immunostained following the same immunofluorescence protocol used for cryostat sections, with the only difference that a brief centrifugation (5 min at 170 $g$) was performed between each solution change to recover the cells at the bottom of Eppendorf tubes and discard the supernatant. Cells were analyzed with BD LSRFortessa and FACSDiva software (Becton Dickinson) on the basis of 10,000 total events (debris excluded).

### RNA-Seq

Total RNA was extracted using TRIzol reagent (Invitrogen), and its integrity was analyzed by using the DNF-471 Standard Sensitivity RNA Analysis Kit on Fragment Analyzer instrument (Advanced Analytical Technology, Ankeny, IA, USA). *RNA-Seq* libraries were prepared from total *RNA* using *TruSeq RNA* Sample Preparation v2 (Illumina, San Diego, CA, USA) according to the manufacturer's protocol and sequenced on an Illumina *NextSeq 500* platform (Illumina). Sequencing reads were trimmed out of the low-quality bases with *FASTX-Toolkit* and mapped on mm9 genome assembly by using *TopHat v2.0.6* (Johns Hopkins University, Baltimore, MD, USA). Differentially expressed genes (DEG) were called by using DESeq2 software (Love *et al*, 2014) (see Appendix Table S1 for complete list of DEG genes). Genes with a $P < 0.01$ were considered for downstream Gene Ontology (GO) analysis (see Appendix Tables S2–S4 for GO analysis with DAVID web software, Ingenuity pathway analysis, or Panther Overrepresentation test, respectively). GO analysis showed in Fig 5O was performed by using DAVID web software (Huang *et al*, 2009).

### Statistics

All data were statistically analyzed and graphically represented using Microsoft Office Excel software and GraphPad Prism (version 7.00). Quantitative data are shown as the mean ± standard error (SEM). For cell percentage/number quantification after immunofluorescence (IF), measurements were performed on at least 9 sections coming from 3 to 5 different animals, unless otherwise stated (see Appendix Table S5 for details). To minimize subjective bias, sample identity (e.g., genotypes) was randomized by associating an identification number to each sample before processing. Fixed embryos with damaged tissues were excluded from any further analysis/processing. Microscope images were processed with Photoshop or ImageJ software, by randomly overlapping fixed-width (100 μm) rectangular boxes on the area of interest (e.g., the lateral pallium of the neocortex), then quantifying positive cells inside the boxes. When calculating percentages over the total cell number, the latter was quantified by counting DAPI$^+$ nuclei, unless otherwise specified. Data were compared by two-tailed Student's $t$-test (when comparing two data groups) or by two-way ANOVA (analysis of variance; for comparison of three or more groups), and statistical significance was set as follows: $*P \leq 0.05$; $**P \leq 0.01$; $***P \leq 0.001$. Sample size, statistical test used, and detailed list of statistical results for each experiment are listed in Appendix Table S5.

### Real-time RT–PCR

Neocortices were dissected in ice-cold PBS1X and stored at $-80°C$. Total RNA was extracted with NucleoSpin RNA II columns (Macherey-Nagel). Analysis was performed as previously described (Bertacchi *et al*, 2013, 2015a,b). Primer sequences are listed in Appendix Table S6.

## Data availability

The *RNA-Seq* datasets produced in this study are available in the following database: Gene Expression Omnibus GSE146595 (https://www.ncbi.nlm.nih.gov/geo/query/acc.cgi?acc = GSE146595).

**Expanded View** for this article is available online.

## Acknowledgements

We are grateful to E. Setti for technical help and F. Calegari for insightful comments. We thank E. Marti and G. Le Dréau for the *Sox2-GFP* and *Tis21-RFP* plasmids, A. Mansouri for the *Pax6 null* line, and Salsabiel El Nagar for cell cycle primers. We also thank the iBV animal facility and especially A. Martres and K. Moneret for animal handling and care, the PRISM Microscopy facility for their regular support, and V. Virolle at the molecular biology facility for the generation of the *PX458 αNr2f1 CRISPR/Cas9* plasmid. Cryostat sections of human samples were kindly provided by C. Allet and P. Giacobini (Lille, France). This work was supported by an ERA-NET Neuron II grant (ImprovVision) ANR-15-NEUR-0002-04, by the Jerome Lejeune Foundation (grant No. 199162), and by "Investments for the Future" LabEx SIGNALIFE (grant ANR-11-LABX-0028-01) to MS and by a postdoctoral fellowship from the city of Nice, France ("Aides Individuelles aux Jeunes Chercheurs"), to MB. The visit of MB in the Lab of SC to perform human organoid experiments was supported by an EMBO short-term fellowship (STF#8035).

## Author contributions

MS and MB conceived and designed the research project. MB, ALR, AL, CF, and RDG performed experiments and collected data. MB, ALR, AL, and MS analyzed data. RDG and SC provided experimental material and contributed to data analysis and discussion. FTM-T, MW, PKK, LF, PK, CP, LP, AG, FD, AS, and CF were involved in the clinical investigation and identification of BBSOAS patients. MS, CF, FTM-T, and LD'I supervised patient data collection and analyzed MRI scans. FN and SO performed and analyzed the *RNA-Seq* transcriptomics. MB prepared manuscript figures. MB and MS wrote the manuscript. All authors provided critical review of results and approved the manuscript.

## Conflict of interest

The authors declare that they have no conflict of interest.

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
