## [Review Process File · The EMBO Journal]

NR2F1 regulates regional progenitor dynamics in the mouse neocortex and cortical gyrification in BBSOAS patients

Michele Bertacchi, Anna Lisa Romano, Agnès Loubat, Frederic Tran Mau-Them, Marjolaine Willems, Laurence Faivre, Philippe Khau van Kien, Laurence Perrin, Françoise Devillard, Arthur Sorlin, Paul Kuentz, Christophe Philippe, Aurore Garde, Francesco Neri, Rossella Di Giaimo, Salvatore Oliviero, Silvia Cappello, Ludovico D'Incerti, Carolina Frassoni and Michèle Studer

Review timeline:

Submission date:	2nd Dec 2019
Editorial Decision:	13th Jan 2020
Revision received:	6th Mar 2020
Editorial Decision:	24th Mar 2020
Revision received:	1st April 2020
Accepted:	15th Apr 2020

Editor: Ieva Gailite

Transaction Report:

1st Editorial Decision

13th Jan 2020

Thank you for submitting your manuscript for consideration by the EMBO Journal. I apologise for the protracted review process due to delays in review submission over the holiday period. We have now received three referee reports on your manuscript, which are included below for your information.

As you will see from the comments, all reviewers appreciate the presented insight into the role of NR2F1 in regulation of cortical neurogenesis and folding. However, reviewers #1 and #2 also raise a number of substantial concerns regarding core aspects of the study that need to be conclusively addressed before they can support publication here. Furthermore, both reviewers indicate that the downstream mechanisms of NR2F1 action have already been described in other publications, and addressing point 8 by reviewer #2 would be useful in expanding the insight into the downstream targets of NR2F1 in this context.

Based on the overall interest expressed in the reports, I would like to invite you to submit a revised version of your manuscript, in which you address the comments of reviewer #1 and #2. I should add that it is The EMBO Journal policy to allow only a single major round of revision and that it is therefore important to resolve the main concerns at this stage. Please contact me if you would like to discuss feasibility of any of the requested experiments.

REFeree REPORTS:

Referee #1:

In this manuscript, Bertacchi et al. investigated the function of NR2F1, a gene whose mutations

cause Boonstra-Bosch-Schaff Optic Atrophy (BBSOA) syndrome in humans. Through detailed phenotyping of six novel patients, they found that in addition to key diagnostic morphological features of BBSOA, they also display regional brain malformation defects. Further studies showed that NR2F1 regionally controls self-renewal and differentiation of neural progenitor cells via regulation of Pax6 and cell cycle genes in mouse models and human cerebral organoids.

General comments:

Overall, I find the data relatively solid. The novelty lies in establishment of new morphological features that expand the BBSOA clinical disease phenotype. Moreover, they characterized in detail the function of NR2F1 in regional regulation of cortical neurogenesis. Specifically, they expand on previous work to systematically evaluate the function of NR2F1 in different neural progenitor populations and across multiple developmental time points, from early fetal to postnatal stages. Regarding molecular mechanisms, however, it is worth noting that previous work by John Rubenstein and colleagues already established that NR2F1 is upstream of PAX6 and regulates cell cycle in neural progenitors (Faedo et al, 2008). This pioneer work should be properly cited.

However, there are multiple aspects of the manuscript that need to be substantially improved. The biggest problem is with statistics. There is no information about replicate numbers at all for any of the experiments. Student t-tests have been used throughout with no information about whether the population data approximate a normal distribution. In experiments where more than one variable is analyzed (such as time and genotype together) a linear model or two-way ANOVA should be used instead of a t-test. There is no information about multiple comparison correction to properly control for type I errors. Besides, the manuscript is poorly written and has not been carefully proofread. There are even cases where conclusions are made without any data or statistical test (see specific points for details). These errors have to be corrected. Other comments and suggestions to improve the manuscript are listed below.

Specific comments:

1. The missense mutations in patients 1 and 5 should be evaluated to establish pathogenicity. This could be done by luciferase assays as have been performed previously using the NR2F1-activated promoter (Chen et al, 2016).
2. On page 7 and Figure 1, the authors claim that NR2F1 expression shows a clear latero-posterior high to medio-anterior low expression gradient. However, Figure 1H-I only shows an anterior-posterior distribution. No latero-medial information is provided.
3. On page 7, the authors claim that "Cortical folding is mainly achieved thanks to a specialized progenitor cell type, called outer Radial Glia or oRG...". However, it has been demonstrated that oRG is not sufficient for gyrification (Kelava et al, 2012; García-Moreno et al, 2012). "Possibly contribute" would be a better term to use.
4. In Figure 1J-M, the authors show that NR2F1 levels are higher in folded regions. The level of NR2F1 is quantified by pixel intensity. Are NR2F1 levels higher because there are more NR2F1 positive progenitors in the folded regions or because individual progenitor cells express higher NR2F1 levels or both? This needs to be clarified.
5. Have the authors characterized Nr2f1 heterozygous mice? Do they show similar phenotypes as homozygous knockouts? This is critical as human patients carry heterozygous mutations.
6. In Figure 3 and EV3, other than volume, does altered progenitor amplification and delayed neurogenesis also occur in the anterior cortex?
7. The labels of Figure EV4 are incorrect. In Figure EV4 panel G, no statistic tests were performed to support "similar results". In Figure EV4 panel H-K, no SATB2 quantification is shown and no data is shown to support the claim that "both early produced Tbr1+ and later generated Satb2+ neurons were over-represented in mutant brains". The data in Figure 4L-N do not support the claim that "cells produced at E17.5 still included neurons at the expense of astrocytes in the mutant."
8. In Figure EV5, why choose different time points for different experiments (E12.5 vs E17.5)?
9. In Figure EV6, Figure 4B-D, Figure EV7 A-F, there are no statistics to support the conclusions.
10. In Figure 6, to demonstrate that PAX6 is downstream of NR2F1, it would be helpful to perform epistasis analysis using Nr2f1^{-/-}; Pax6^{-/-} neurospheres. If the effect of Nr2f1 on progenitor growth is solely dependent on Pax6, the phenotype of double knockout should phenocopy Pax6^{-/-}.
11. In Figure 7, the authors could also perform NR2F1; PAX6 double knockout to determine if Pax6 is also a downstream effector of NR2F1 in brain organoids.
12. SP8 and NR2F1 have been shown to display a reciprocal expression pattern. Is the expression of SP8 also changed in mutants? Will that also contribute to the phenotype? It would be helpful to

discuss this relationship (Borello et al, 2014).

13. On page 22, the authors write "... low levels of NR2F1 would lead to accelerated cell cycle progression and altered regulation of oRG-dependent neurogenesis...". There is very little data characterizing oRG-dependent neurogenesis in this manuscript (Figure EV7). Much more work is needed to support this conclusion. I would just delete this from the manuscript.

Reference:

- Borello U, Madhavan M, Vilinsky I, Faedo A, Pierani A, Rubenstein J & Campbell K (2014) Sp8 and COUP-TF1 reciprocally regulate patterning and fgf signaling in cortical progenitors. *Cereb. Cortex* 24: 1409-1421
- Chen CA, Bosch DGM, Cho MT, Rosenfeld JA, Shinawi M, Lewis RA, Mann J, Jayakar P, Payne K, Walsh L, Moss T, Schreiber A, Schoonveld C, Monaghan KG, Elmslie F, Douglas G, Boonstra FN, Millan F, Cremers FPM, McKnight D, et al (2016) The expanding clinical phenotype of Bosch-Boonstra-Schaaf optic atrophy syndrome: 20 new cases and possible genotype-phenotype correlations. *Genet. Med.* 18: 1143-1150
- Faedo A, Tomassy GS, Ruan Y, Teichmann H, Krauss S, Pleasure SJ, Tsai SY, Tsai MJ, Studer M & Rubenstein JLR (2008) COUP-TFI coordinates cortical patterning, neurogenesis, and laminar fate and modulates MAPK/ERK, AKT, and β -catenin signaling. *Cereb. Cortex* 18: 2117-2131
- García-Moreno F, Vasistha NA, Trevia N, Bourne JA & Molnár Z (2012) Compartmentalization of cerebral cortical germinal zones in a lissencephalic primate and gyrencephalic rodent. *Cereb. Cortex* 22: 482-492
- Kelava I, Reillo I, Murayama AY, Kalinka AT, Stenzel D, Tomancak P, Matsuzaki F, Lebrand C, Sasaki E, Schwamborn JC, Okano H, Huttner WB & Borrell V (2012) Abundant occurrence of basal radial glia in the subventricular zone of embryonic neocortex of a lissencephalic primate, the common marmoset *Callithrix jacchus*. *Cereb. Cortex* 22: 469-481

Referee #2:

This is a very nice study by the Studer lab focused on the functional analysis of the transcription factor NR2F1 during cerebral cortex development. The authors begin presenting the identification of a new set of patients with point mutations in this locus that present learning disabilities associated to malformations of cortical folding, primarily polymicrogyria. They show that in human embryo cortex NR2F1 is highly expressed in progenitor cells but, interestingly, at different levels between emerging folds versus fissures. Using mouse as experimental model they perform a very comprehensive analysis of progenitor cell proliferation, brain growth and cell fate, using a wide variety of complementary approaches. They demonstrate that NR2F1 plays an essential role in limiting progenitor cell self-amplification and subsequent neurogenesis. Importantly, they demonstrate that defects in *Nr2f1* null mice are heavily regionalized, in agreement with the regional variations in endogenous expression levels of this gene. Via candidate gene analyses the authors identify several cell cycle genes downregulated in *Nr2f1* mutant mouse embryos, and focus on *Pax6* as a prime target for *Nr2f1*, to then functionally demonstrate that changes in *Pax6* expression levels mediate some of the effects of *Nr2f1* on cortical progenitor cells, including cell cycle duration, fate of cell divisions and self-renewal capacity. Finally, the study closes the circle going back to human cerebral organoids to show the same reciprocal levels of expression between *Nr2f1* and *Pax6*, and that NR2F1 also favors progenitor cell amplification over neurogenesis in human. The study is remarkably exhaustive, analyzing many different parameters, usually in several independent ways, with a sound technical performance of experiments and analyses, and the conclusions are definitely justified by the findings. In fact, there are some supplementary figures that include so many interesting and important results that deserve being main figures, namely EV4 and EV6. This is a very well performed study addressing the fundamentally important issue of the developmental origins of polymicrogyria, one of the most enigmatic and poorly understood malformations of human brain development with profound effects on learning disabilities, cognition and epilepsy. For these reasons, I definitely recommend publication of this study, upon addressing the following concerns:

Major issues:

- 1- A main conclusion of this study is that loss of *Nr2f1* leads to increase progenitor cell self-

amplification, followed at later stages by greater neuron production and a larger cortex. This process includes an outstanding increase in the generation of bRGCs (or oRGs) in *Nr2f1* knock-out mouse embryos, as shown in multiple analyses: Figures 3, 4 EV3, EV5, EV7. However, this seems to contradict the initial observations that in the normally developing human embryo low levels of NR2F1 expression are found in sulci, with very few bRGCs, whereas high NR2F1 expression corresponds to gyri, with very abundant bRGCs and neurogenesis.

2- The pattern of NR2F1 stain is well demonstrated for the anterior-posterior axis, but not for the lateral-medial axis, for which stains must be provided to back-up the quantification.

3- In Fig 1M, in which layer was pixel intensity measured? It seems that all layers may be similarly different between gyrus and sulcus, but this needs to be specifically measured and specified in the quantification.

4- Related to the previous point, in Fig. 1O cell density varies very significantly between gyrus and sulcus, most particularly HOPX+ bRGCs. Was this quantification of NR2F1+ cells normalized to the total number of HOPX+ cells, or total number of DAPI+ cells? Otherwise, this does not reflect differences in cellular NR2F1 expression, but in overall HOPX+ cell density. In fact, from the images in Fig. N', N' it seems that between gyrus and sulcus there are similar densities of NR2F1+ cells, even with a very remarkable difference in HOPX+ cells! If this is so, then the data plotted in O is misleading and the conclusion to be drawn is the opposite: NR2F1 is expressed at high frequency in sulcus even with a very low density of HOPX+ cells. Given this conclusion, the authors should really determine if there are differences in density of NR2F1+ cells between gyrus and sulcus, in addition to the well demonstrated difference in NR2F1 antibody stain intensity.

5- Increased numbers of bRGs is not only the result of amplification of Pax6+ progenitors, but also combined with their delamination and translocation to basal positions, which is clearly also enhanced in NR2F1 mutant mice. What is the mechanism by which this TF regulates such cell behavior? Does the transcriptomic analysis shed any light on this? The authors should look a bit into this, as it is seemingly essential to their PMG phenotype in patients.

6- According to Extended Figure EV4F, the authors find very small differences (if any, depending on age) in cell cycle exit, but they previously describe dramatically increased abundance of Pax6+, Tbr2+, and Pax6+Tbr2+ cells, at E15.5 and later, followed by greater numbers of neurons being generated. How can this be reconciled? This point is confusing and the authors should at least discuss it in their manuscript. For example, cell cycle exit at E16.5 should be much higher in mutant embryos, when the authors find a high degree of progenitor consumption in controls but high amplification and generation of IPCs in mutants (Fig. 3M).

7- The general delay in cortex development parameters observed in these mutants seems to contradict the normal situation, where a rostro-caudal time-lag in cortex development anti-correlates with endogenous levels of NR2F1 expression. That is, occipital cortex with high levels of NR2F1 has a delayed development compared to rostral cortex with very low levels of this gene. It seems that depletion of NR2F1 should accelerate differentiation of the caudal cortex, not the opposite as found in this analysis. This is worth discussing.

8- To gain insights into the transcriptional mechanisms of action of *Nr2f1*, the authors nicely perform pRT-PCR on a set of candidate cell cycle genes comparing between control and knock-out mouse embryos. But only three of their 20 candidates show significant changes, in addition to Pax6. The study demonstrates that these mutant embryos have multiple deficits in addition to changes in cell cycle, including bRG generation, amplification of progenitors, delay in neurogenesis, etc., which are complex processes involving multiple gene networks. It would be much more informative to have a full transcriptomic profile analysis of these mutants, which would likely provide a more comprehensive picture of the genetic changes and signaling cascades perturbed by loss of *Nr2f1* and underlying the complex phenotype of these mutants.

9- Throughout the manuscript, the authors refer to basal Radial Glia cells as outer Radial Glia (oRG). These two names are being used interchangeably since the discovery of these cells, but outer Radial Glia is really a misleading misnomer, as these cells are found in both outer SVZ (OSVZ) and inner SVZ (ISVZ), and the alternative cell type are not called inner Radial Glia cells. Instead, under

the perspective of cell biology it makes most sense to call them basal and apical RGCs, respectively (bRGCs, aRGCs), as classical Radial Glia cells divide apically and those in OSVZ and ISVZ divide basally. The authors should adopt this consensus terminology throughout the manuscript.

Minor issues

1- In Figure EV5, why is cell cycle duration expressed in arbitrary units? This should be in hours.

2- In the analysis of cell cycle length, legend to Figure 4H indicates that "y-intercept is proportional to S-phase length". This is only correct if total cell cycle length remains constant. Using this method, S-phase length corresponds to the x-intercept, and this depends on the y-intercept combined with the slope of the fit line. Two fit lines may have the same x-intercept (same T_s in hours) but different y-intercept.

3- Several quantifications throughout the study do not indicate statistical significance, which must be fixed. This is the case for Figure panels 1I, 1M, 1O, 3U, 4B-C, 6A, 6B, 6E, 6L, 7C and 7J. The number of units analyzed (cells, neurospheres, embryos,...) must also be indicated everywhere.

4- In Figure 6L, the authors must indicate the range of division plane angles that were considered as vertical, oblique or horizontal. In fact, a representation of the exact values for each cell, and analysis of the population distribution, is preferable.

Referee #3:

Bertacchi and coworkers report on altered cortical progenitor development in NR2F1 conditional knockout mice, a mouse model of a neurodevelopmental disease referred to as BBSOA (Boonstra-Bosch-Schaff Optic Atrophy). As an entry to the story, the authors present MRI pictures of BBSOA patient brains with NR2F1 haploinsufficiencies presenting abnormal gyrifications. They also show differential expression of NR2F1 in prospective sulci and gyri in developing human cortex. The authors then switch to the NR2F1 conditional knockout mice and find that early and complete loss of NR2F1 causes cell cycle acceleration, increased neural progenitor self-renewal and sustained Pax6 expression, causing a delay in neurogenesis and lateral expansion of the posterior hemispheres. Consequently, the expanded progenitor pool produces an enlarged outer radial glia and neuronal output, leading to the radial expansion of the smooth, unfolded cortex. Additional support for NR2F1 regulating progenitor physiology came from experiments with human brain organoids. The authors propose NR2F1 as a novel regulator of neural progenitor proliferation and human cortical folding.

This manuscript contains some interesting and novel observations on NR2F1 functions in the developing cortex. The data are of very high technical quality and the results are conclusive and convincing. This reviewer does not have any specific suggestions on how to improve the experiments and/or data analysis. My enthusiasm is a bit dampened by the lack of connection between the possible roles of NR2F1 in cortical gyrification based on human MRI data and in regulating progenitor proliferation based on mouse knockout data. Whether these two functions are linked remains to be demonstrated. The authors present possible ways to proceed in the discussion of the manuscript.

Please see next page.

RESPONSES TO REFEREES MS N° EMBOJ-2019-104163

Referee #1:

General comments:

Overall, I find the data relatively solid. The novelty lies in establishment of new morphological features that expand the BBSOA clinical disease phenotype. Moreover, they characterized in detail the function of NR2F1 in regional regulation of cortical neurogenesis. Specifically, they expand on previous work to systematically evaluate the function of NR2F1 in different neural progenitor populations and across multiple developmental time points, from early fetal to postnatal stages.

Authors: We thank the Reviewer for acknowledging the importance and potential of our study.

Regarding molecular mechanisms, however, it is worth noting that previous work by John Rubenstein and colleagues already established that NR2F1 is upstream of PAX6 and regulates cell cycle in neural progenitors (Faedo et al, 2008). This pioneer work should be properly cited.

Authors: We apologize with the reviewer for not having properly cited the Faedo et al. 2008 paper while mentioning Pax6. We made sure to add the paper in additional places, such as in the results section while describing Fig. 5 when we state that "Pax6 levels can be modulated by Nr2f1". However, since this paper did not show any evidence of direct regulation of Nr2f1 on Pax6 expression, we also added the work of Tang et al., 2010, where the Nr2f1-mediated control of Pax6 expression is demonstrated to be direct, at least in retinal tissue.

However, there are multiple aspects of the manuscript that need to be substantially improved. The biggest problem is with statistics. There is no information about replicate numbers at all for any of the experiments. Student t-tests have been used throughout with no information about whether the population data approximate a normal distribution. In experiments where more than one variable is analyzed (such as time and genotype together) a linear model or two-way ANOVA should be used instead of a t-test. There is no information about multiple comparison correction to properly control for type I errors.

Authors: We deeply apologize for the lack of enough information regarding statistics. We have now substantially improved our statistical analysis by using GraphPad and double checked all our data. Two-way ANOVA has been used when analyzing more than one variable, and we kept the Student t-test for comparisons of two conditions. All figures have now been updated with the new results. To avoid adding all the statistics in the text or in figure legends, we have now formatted a new supplementary table (Appendix Table S5) indicating the sample size (n), the statistical tests used in the different figures and the results of the statistical output for all images presented in the manuscript.

Besides, the manuscript is poorly written and has not been carefully proofread. There are even cases where conclusions are made without any data or statistical test (see specific points for details). These errors have to be corrected. Other comments and suggestions to improve the manuscript are listed below.

Authors: We apologize with the reviewer of this lack of statistical analysis. All data have now been now statistically tested, so that any statement is correctly corroborated by data analysis (see also above). We have also carefully proofread the

manuscript to correct any typos and/or mistakes.

Specific comments:

1. The missense mutations in patients 1 and 5 should be evaluated to establish pathogenicity. This could be done by luciferase assays as have been performed previously using the NR2F1-activated promoter (Chen et al, 2016).

Authors: We agree with the reviewer that additional experiments are necessary to further explore the pathogenic effect of all novel NR2F1 variants. To this aim, we are at present setting up an innovative molecular approach, based on genetic code expansion (GCE) and 3D protein modelling, which will evaluate how NR2F1 missense variants impact protein structure and its molecular function. We will test the majority of reported and unreported NR2F1 missense mutations for their ability either to transactivate a target gene or to dimerize with partners. This will help us to correlate genotype to phenotype in patient groups and unveil how the protein structure can affect its function in the presence of representative disease mutations. The results, which will also contribute in understanding the phenotypic variability in severity among patients, will be reported elsewhere.

The novel mutations found in patients 1 and 5, and particularly in patient 1, alter a hotspot sequence that has already been tested by a luciferase assay in Chen et al., 2016. Moreover, we believe that the *in silico* predictions of point variant pathogenesis that human geneticists have performed, already denote a first step in predicting variant pathogenicity.

2. On page 7 and Figure 1, the authors claim that NR2F1 expression shows a clear latero-posterior high to medio-anterior low expression gradient. However, Figure 1H-I only shows an anterior-posterior distribution. No latero-medial information is provided.

Authors: The reviewer is correct. Because of space constraint in Figure 1, we have now added images of latero-medial NR2F1 expression in EV1 G'-G'''.

3. On page 7, the authors claim that "Cortical folding is mainly achieved thanks to a specialized progenitor cell type, called outer Radial Glia or oRG...". However, it has been demonstrated that oRG is not sufficient for gyrification (Kelava et al, 2012; García-Moreno et al, 2012). "Possibly contribute" would be a better term to use.

Authors: Sorry for this oversight. We have now amended the text and added the suggested references.

4. In Figure 1J-M, the authors show that NR2F1 levels are higher in folded regions. The level of NR2F1 is quantified by pixel intensity. Are NR2F1 levels higher because there are more NR2F1 positive progenitors in the folded regions or because individual progenitor cells express higher NR2F1 levels or both? This needs to be clarified.

Authors: We thank the Reviewer for pointing out this problem. We agree that local NR2F1 levels could also depend on different cell densities between folds and sulci. To avoid any confusion, we repeated the pixel intensity quantification using ImageJ on single nuclei in high magnification images (note different expression levels in Fig.1 K', L'). We performed measurements on VZ aRGs (Fig 1M), SVZ TBR2+ IPs (Fig 1O), oSVZ bRGs (Fig 1Q) and CP neurons (Fig 1M'). Images in Fig. 1 and EV1 have now been improved to show the new data. When quantified at a single cell level (hence independently of cell number/density), expression levels of NR2F1 in progenitors still depict a difference between sulci and gyri.

5. Have the authors characterized Nr2f1 heterozygous mice? Do they show similar phenotypes as homozygous knockouts? This is critical as human patients carry heterozygous mutations.

Authors: While we believe that homozygous *KO* embryos can better highlight cortical defects, we totally agree with the Reviewer that *HET* mice are of great interest, since they better recapitulate BBSOA patients haploinsufficient for NR2F1. In this revised version, we added new data of key experiments on *HET* embryos, such as the effect of reduced Nr2f1 dosage on: altered ratio among different NP subtypes (new Expanded Figure EV3A-F), abnormal progenitor differentiation index (new Expanded Figure EV3G-J), cell cycle time alteration (Expanded Figure 5), and increased Pax6 expression (new Appendix Figure S4). The results of these additional experiments on heterozygous mutants go in the same direction to homozygous *KO* embryos, even if the differences between *WT* and *HET* embryos are less pronounced than between *WT* and *KO*. Figures and text have been modified accordingly, in order to integrate the new data.

6. In Figure 3 and EV3, other than volume, does altered progenitor amplification and delayed neurogenesis also occur in the anterior cortex?

Authors: Our data show that Nr2f1 can play different roles in NP regulation depending on the A-P axis. In the caudal cortex where the gene is expressed at its highest levels, its loss produces the most severe effect. Our data fail to show any neurogenesis defect in the anterior/motor cortex; however, we are planning to build a mathematical model predicting the rate of neurogenesis based on distinct parameters, such as cell cycle length, Q index and number of progenitors along the A-P and L-M axes of control, *HET* and *HOM* embryos at different developmental ages.

7. The labels of Figure EV4 are incorrect. In Figure EV4 panel G, no statistic tests were performed to support "similar results". In Figure EV4 panel H-K, no SATB2 quantification is shown and no data is shown to support the claim that "both early produced Tbr1+ and later generated Satb2+ neurons were over-represented in mutant brains". The data in Figure 4L-N do not support the claim that "cells produced at E17.5 still included neurons at the expense of astrocytes in the mutant."

Authors: We deeply apologize with the reviewer that we did not add any statistics on the panels mentioned above. Figure EV4 was carefully revised and proper statistical analysis was added. For clarity, we also decided to move the short term differentiation assay to the main figure 3 (Fig 3Q-R'), we substituted Satb2 staining with Cux1 staining (Fig EV4G-G''), a more suitable marker for superficial neurons, and finally we improved statistical representations in Fig EV4A,D,E,F'',G'',J,L'' and M. A careful quantification of Ctip2+, Satb2+ and Tbr1+ neurons in P8 cortices was also added (Fig EV4Q) to support the claim that Tbr1+ and Satb2+ neurons are over-represented in mutant brains. Concerning gliogenesis, we changed the statement as follows: "cells produced at E17.5 still included a high number of neurons in mutant brains together with slightly increased gliogenesis", (see Fig EV4J). We hope the reviewer will agree with us that the figure is clearer in its present form. Importantly, any statement in the text is now correctly linked to appropriate statistical analysis.

8. In Figure EV5, why choose different time points for different experiments (E12.5 vs E17.5)?

Authors: Sorry for the confusion. Regarding Figure EV5, we chose E17.5 as the stage when basal progenitors are the most abundant (especially in *KO* embryos) to better appreciate possible differences in PH3⁺ NP populations along the A-P axis. For cell cycle analysis, on the contrary, we decided to show an earlier time point (E12.5) when progenitors are highly cycling, as no differences in cell cycle duration can be found at later stages (after E15.5; see Fig 4E). We now clearly state in the text the reasons for choosing these specific time points.

9. In Figure EV6, Figure 4B-D, Figure EV7 A-F, there are no statistics to support the conclusions.

Authors: As mentioned above, we have now performed proper statistics in all our experiments and reported the significance to all our graphs. For Figure 4B-D, statistical analysis is reported in Appendix Table S5.

10. In Figure 6, to demonstrate that PAX6 is downstream of NR2F1, it would be helpful to perform epistasis analysis using *Nr2f1*^{-/-}; *Pax6*^{-/-} neurospheres. If the effect of *Nr2f1* on progenitor growth is solely dependent on *Pax6*, the phenotype of double knockout should phenocopy *Pax6*^{-/-}.

Authors: Our experiments show that *Pax6* is slightly upregulated in neurospheres and in embryos (33% increase by qPCR). The rationale of our double mutant experiments (*Nr2f1*^{-/-}; *Pax6*^{-/+}) is to rescue the phenotype by diminishing *Pax6* expression to a level as similar as possible to what is physiological. By using a double KO (*Nr2f1*^{-/-}; *Pax6*^{-/-}), we will induce a supplementary phenotype since complete loss of *Pax6* produces additional defects independent of *Nr2f1*.

11. In Figure7, the authors could also perform NR2F1; PAX6 double knockout to determine if *Pax6* is also a downstream effector of NR2F1 in brain organoids.

Authors: Please, see the response above. We noticed that double KO will not rescue the phenotype (which was instead our aim) but induces additional defects. Here, we aimed to demonstrate that decreased levels of *Pax6* upon *Nr2f1* loss would rescue some of the new phenotypes we presented in our study. Although we agree with the reviewer that further studies would be necessary to confirm a direct regulation of *Nr2f1* on *Pax6* in the brain, a previous paper of Tang et al., 2010 already showed a direct regulation of *Nr2f1* on *Pax6* in the eye.

12. SP8 and NR2F1 have been shown to display a reciprocal expression pattern. Is the expression of SP8 also changed in mutants? Will that also contribute to the phenotype? It would be helpful to discuss this relationship (Borello et al, 2014).

Authors: We thank the reviewer for the suggestion. The relationship between Sp8 and *Nr2f1* has already been demonstrated in Borello et al., 2014, as mentioned by the reviewer, but also in Alfano et al., 2014, in which we show by qPCR and in situ hybridization that Sp8 transcript is indeed increased in *Nr2f1* conditional mutants. Nevertheless, we performed a new antibody Sp8 staining on E12.5 embryos and confirmed a reciprocal inhibition between Sp8 and *Nr2f1* at the protein level (see image below). In *Nr2f1* KO embryos, Sp8 is up-regulated, notably in the dorsal pallium and particularly in ganglionic eminences, as previously shown in Touzot et al., 2016. However, since Sp8 is not particularly changed in regions of interest of this study, i.e., the lateral-most pallium, we do not believe that upregulation of Sp8 can contribute to the phenotype and, thus decided not to insert these data in our Figures.

13. On page 22, the authors write "... low levels of NR2F1 would lead to accelerated cell cycle progression and altered regulation of oRG-dependent neurogenesis...". There is very little data characterizing oRG-dependent neurogenesis in this manuscript (Figure EV7). Much more work is needed to support this conclusion. I would just delete this from the manuscript.

Authors: We agree with the reviewer and substituted the sentence with "and altered ratio of specific NP subtypes", to better highlight our findings.

Reference:

Touzot, A., Ruiz-Reig, N., Vitalis, T. & Studer, M. Molecular control of two novel migratory paths for CGE-derived interneurons in the developing mouse brain. *Development* 143, 1753–1765 (2016).

Alfano, C., Magrinelli, E., Harb, K., Hevner, R. F. & Studer, M. Postmitotic control of sensory area specification during neocortical development. *Nat. Commun.* 5, (2014).

Borello U, Madhavan M, Vilinsky I, Faedo A, Pierani A, Rubenstein J & Campbell K (2014) Sp8 and COUP-TF1 reciprocally regulate patterning and fgf signaling in cortical progenitors. *Cereb. Cortex* 24: 1409-1421

Chen CA, Bosch DGM, Cho MT, Rosenfeld JA, Shinawi M, Lewis RA, Mann J, Jayakar P, Payne K, Walsh L, Moss T, Schreiber A, Schoonveld C, Monaghan KG, Elmslie F, Douglas G, Boonstra FN, Millan F, Cremers FPM, McKnight D, et al (2016) The expanding clinical phenotype of Bosch-Boonstra-Schaaf optic atrophy syndrome: 20 new cases and possible genotype-phenotype correlations. *Genet. Med.* 18: 1143-1150

Faedo A, Tomassy GS, Ruan Y, Teichmann H, Krauss S, Pleasure SJ, Tsai SY, Tsai MJ, Studer M & Rubenstein JLR (2008) COUP-TFI coordinates cortical patterning, neurogenesis, and laminar fate and modulates MAPK/ERK, AKT, and β -catenin signaling. *Cereb. Cortex* 18: 2117-2131

García-Moreno F, Vasistha NA, Trevia N, Bourne JA & Molnár Z (2012) Compartmentalization of cerebral cortical germinal zones in a lissencephalic primate and gyrencephalic rodent. *Cereb. Cortex* 22: 482-492

Kelava I, Reillo I, Murayama AY, Kalinka AT, Stenzel D, Tomancak P, Matsuzaki F, Lebrand C, Sasaki E, Schwamborn JC, Okano H, Huttner WB & Borrell V (2012)

Abundant occurrence of basal radial glia in the subventricular zone of embryonic neocortex of a lissencephalic primate, the common marmoset callithrix jacchus. Cereb. Cortex 22: 469-481

Referee #2:

.....The study is remarkably exhaustive, analyzing many different parameters, usually in several independent ways, with a sound technical performance of experiments and analyses, and the conclusions are definitely justified by the findings. In fact, there are some supplementary figures that include so many interesting and important results that deserve being main figures, namely EV4 and EV6.

Authors: We thank the Reviewer for the very positive and encouraging remarks regarding our study. Following the reviewer's suggestion, we have now moved some images from the EdU short term injection experiments (previously EV4A, B) to main Figure 3 (now Fig 3Q-R'), as we believe they nicely complement the symmetric/asymmetric short-term in vivo assay (Fig 3M-P). Both assays suggest a delay in neurogenesis. Unfortunately, we could not move so many data to the main figures; therefore, we kept them in Expanded view Figures (EV) and/or Appendix Figures (AF).

This is a very well performed study addressing the fundamentally important issue of the developmental origins of polymicrogyria, one of the most enigmatic and poorly understood malformations of human brain development with profound effects on learning disabilities, cognition and epilepsy. For these reasons, I definitely recommend publication of this study, upon addressing the following concerns:

Major issues:

1- A main conclusion of this study is that loss of Nr2f1 leads to increase progenitor cell self-amplification, followed at later stages by greater neuron production and a larger cortex. This process includes an outstanding increase in the generation of bRGCs (or oRGs) in Nr2f1 knock-out mouse embryos, as shown in multiple analyses: Figures 3, 4 EV3, EV5, EV7. However, this seems to contradict the initial observations that in the normally developing human embryo low levels of NR2F1 expression are found in sulci, with very few bRGCs, whereas high NR2F1 expression corresponds to gyri, with very abundant bRGCs and neurogenesis.

Authors: We thank the Reviewer for raising this point, which surely needs further discussion. It can be confusing to compare two different contexts: on one side the significance of different expression levels of human NR2F1 along cortical folds; on the other side, the effects of early and complete loss of Nr2f1 in a mouse model. Our present and previously published data support a role of Nr2f1 in promoting neurogenesis; thus, loss of Nr2f1 will delay neurogenesis, ultimately resulting in an expansion of Pax6+ radial glia cells (apical and basal). In humans, the medium-high NR2F1 expression levels in cortical folds correlate with the presence of high numbers of basal progenitors and neurons, i.e. where neurogenesis is highest, whereas low NR2F1 levels in sulci is associated with low neurogenesis. Thus, in both cases the data converge to the model that Nr2f1/NR2F1 is a pro-neurogenic factor whose role is promoting proliferating neurogenic progenitors to produce high numbers of cortical neurons. We have now discussed these apparent controversial observations in the text.

2- The pattern of NR2F1 stain is well demonstrated for the anterior-posterior axis, but

not for the lateral-medial axis, for which stains must be provided to back-up the quantification.

Authors: We fully agree and have now added new images of NR2F1 expression in human GW11 cortices along L-M axis, as also requested by Reviewer 1 (see EV1 G'-G''').

3- In Fig 1M, in which layer was pixel intensity measured? It seems that all layers may be similarly different between gyrus and sulcus, but this needs to be specifically measured and specified in the quantification.

Authors: As also requested by Reviewer 1, we repeated pixel intensity analysis on single cells to better quantify NR2F1 expression level independently of cell density. (Figures 1J-R). We performed measurements on VZ aRGs (Fig 1M), SVZ TBR2+ IPs (Fig 1O), oSVZ bRGs (Fig 1Q) and CP neurons (Fig 1M'). Images in Fig. 1 and EV1 have now been improved to show the new data. When quantified at a single cell level, expression of NR2F1 in progenitors still depicts differences in levels between sulci and gyri.

4- Related to the previous point, in Fig. 1O cell density varies very significantly between gyrus and sulcus, most particularly HOPX+ bRGs. Was this quantification of NR2F1+ cells normalized to the total number of HOPX+ cells, or total number of DAPI+ cells? Otherwise, this does not reflect differences in cellular NR2F1 expression, but in overall HOPX+ cell density. In fact, from the images in Fig. N', N' it seems that between gyrus and sulcus there are similar densities of NR2F1+ cells, even with a very remarkable difference in HOPX+ cells! If this is so, then the data plotted in O is misleading and the conclusion to be drawn is the opposite: NR2F1 is expressed at high frequency in sulcus even with a very low density of HOPX+ cells. Given this conclusion, the authors should really determine if there are differences in density of NR2F1+ cells between gyrus and sulcus, in addition to the well demonstrated difference in NR2F1 antibody stain intensity.

Authors: We agree with the reviewer and apologize for the confusion. The number of double HOPX+NR2F1+ cells per counting area has been quantified to better compare the cell density between sulci and gyri (graph in Figure 1R). However, as also raised by Reviewer 1, it is true that cell density could affect our measurement of cellular NR2F1 expression. For this reason, we repeated the analysis on single cells to quantify NR2F1 expression levels by pixel intensity, independently of local cell density (Figures 1J-Q). Our results confirm increased levels of NR2F1 in gyri compared to sulci in single progenitor cells but not in CP differentiating neurons (Figure 1 J-L').

5- Increased numbers of bRGs is not only the result of amplification of Pax6+ progenitors, but also combined with their delamination and translocation to basal positions, which is clearly also enhanced in NR2F1 mutant mice. What is the mechanism by which this TF regulates such cell behavior? Does the transcriptomic analysis shed any light on this? The authors should look a bit into this, as it is seemingly essential to their PMG phenotype in patients.

Authors: We thank the reviewer for this insightful comment. By looking at our new transcriptomic analysis, requested by this reviewer and presented in Figure 5N-P and Appendix Fig 6, we indeed found as potential Nr2f1 targets some cell adhesion molecules, which could impinge on changes in cell behavior and delamination. They have been listed in Appendix Tables 2-4. We have now mentioned these new findings in the text, but we plan to study them in more details in future studies.

6- According to Extended Figure EV4F, the authors find very small differences (if any, depending on age) in cell cycle exit, but they previously describe dramatically increased abundance of Pax6+, Tbr2+, and Pax6+Tbr2+ cells, at E15.5 and later, followed by greater numbers of neurons being generated. How can this be reconciled? This point is confusing and the authors should at least discuss it in their manuscript. For example, cell cycle exit at E16.5 should be much higher in mutant embryos, when the authors find a high degree of progenitor consumption in controls but high amplification and generation of IPCs in mutants (Fig. 3M).

Authors: We apologize for this confusion and have tried to amend Figure EV4 to make it more comprehensible. Regarding the short-term differentiation assay, consisting in Ki67 staining 24 hours after EdU injection (previously EV4 E,F, now EV4 A,D), we decided to keep three key ages representing early, middle and late developmental stages (E13.5, E16.5 and P0). In these graphs, we show data of the number of differentiating cells (EV4A), or the percentage of cells exiting the cell cycle over the total EdU+ population (Q index; EV4D). Both approaches support a delay in neurogenesis at E13.5 (EV4A,D), followed by a boost of neuronal differentiation at later time points (EV4A). However, this increased trend to differentiate at later stages (E16.5 and P0) is no longer evident when normalizing the differentiating EdU+Ki67-cells on the total EdU+ population (EV4D), showing that the tendency of NPs to exit the cell cycle in mutant brains at later time points is not changed. Hence, the increased NP populations (Fig 3A-I) and the higher neuronal production (Figure EV4A) observed in KOs could be due to an indirect consequence of a faster cell cycle (as shown in Figure 4) and delayed neurogenesis (Figure 3M-R'), rather than to a high tendency of exiting the cell cycle. Together, our data suggest that increased neuronal production is the result of prolonged NP self-renewal and faster cell cycle, and not of an increase in cell cycle exit that would, on the contrary, result in early NP pool exhaustion.

7- The general delay in cortex development parameters observed in these mutants seems to contradict the normal situation, where a rostro-caudal time-lag in cortex development anti-correlates with endogenous levels of NR2F1 expression. That is, occipital cortex with high levels of NR2F1 has a delayed development compared to rostral cortex with very low levels of this gene. It seems that depletion of NR2F1 should accelerate differentiation of the caudal cortex, not the opposite as found in this analysis. This is worth discussing.

Authors: We thank the reviewer for raising this point. We believe that the role of Nr2f1 along the A-P axis is more related to impinge a sensory identity than to regulate neurogenesis. Nevertheless, the regulation of the tangential extent of individual areas is known to be under the control of cell cycle duration and rate of cell-cycle re-entry of progenitors. Thus, we propose that within the occipital cortex, where Nr2f1 levels are very high, the gene has the role to control the size of the progenitor pool and its progeny. We have extensively discussed this issue in the text.

8- To gain insights into the transcriptional mechanisms of action of Nr2f1, the authors nicely perform pRT-PCR on a set of candidate cell cycle genes comparing between control and knock-out mouse embryos. But only three of their 20 candidates show significant changes, in addition to Pax6. The study demonstrates that these mutant embryos have multiple deficits in addition to changes in cell cycle, including bRG generation, amplification of progenitors, delay in neurogenesis, etc., which are complex processes involving multiple gene networks. It would be much more informative to have a full transcriptomic profile analysis of these mutants, which would likely provide a more comprehensive picture of the genetic changes and

signaling cascades perturbed by loss of Nr2f1 and underlying the complex phenotype of these mutants.

Authors: Following the Reviewer's suggestion, we have performed an unbiased analysis of differentially expressed genes (DEGs) by RNA-Seq of *WT versus KO* cortices at E15.5 to explore, in a more comprehensive way, the genetic mis-regulation at a developmental stage when neurogenesis is at its peak. Our results have been integrated in Figure 5 (N-P) and Appendix Figure 6. Our Gene Ontology (GO) analysis confirms that Nr2f1 orchestrates key developmental neurogenesis processes, such as neuron differentiation, but also axonogenesis and cell morphogenesis. Interestingly, the MAPK signaling pathway turns out to be highly significant, as previously mentioned by Faedo et al., 2008. The way Nr2f1 influences all these targets and signaling pathways will be the object of future studies.

9- Throughout the manuscript, the authors refer to basal Radial Glia cells as outer Radial Glia (oRG). These two names are being used interchangeably since the discovery of these cells, but outer Radial Glia is really a misleading misnomer, as these cells are found in both outer SVZ (OSVZ) and inner SVZ (ISVZ), and the alternative cell type are not called inner Radial Glia cells. Instead, under the perspective of cell biology it makes most sense to call them basal and apical RGCs, respectively (bRGCs, aRGCs), as classical Radial Glia cells divide apically and those in OSVZ and ISVZ divide basally. The authors should adopt this consensus terminology throughout the manuscript.

Authors: We agree with the Reviewer and have adopted this terminology in the Text and Figures.

Minor issues

1- In Figure EV5, why is cell cycle duration expressed in arbitrary units? This should be in hours.

Authors: We have now amended the cell cycle duration in hours, as requested (Figure 4E).

2- In the analysis of cell cycle length, legend to Figure 4H indicates that "y-intercept is proportional to S-phase length". This is only correct if total cell cycle length remains constant. Using this method, S-phase length corresponds to the x-intercept, and this depends on the y-intercept combined with the slope of the fit line. Two fit lines may have the same x-intercept (same T_s in hours) but different y-intercept.

Authors: Sorry for the confusion. We amended this in both figure legends and methods.

3- Several quantifications throughout the study do not indicate statistical significance, which must be fixed. This is the case for Figure panels 1I, 1M, 1O, 3U, 4B-C, 6A, 6B, 6E, 6L, 7C and 7J. The number of units analyzed (cells, neurospheres, embryos,...) must also be indicated everywhere.

Authors: As also requested by Reviewer 1, we have now substantially improved our statistical analysis by using GraphPad software and re-test all the data. We created a supplementary table (Appendix Table S5) summarizing the sample size (n), the statistical test used and the results of statistical analysis for all images of the paper.

4- In Figure 6L, the authors must indicate the range of division plane angles that were considered as vertical, oblique or horizontal. In fact, a representation of the exact

values for each cell, and analysis of the population distribution, is preferable.

Authors: Division plane angles are now correctly displayed. Thank you for this remark.

Referee #3:

.....

This manuscript contains some interesting and novel observations on NR2F1 functions in the developing cortex. The data are of very high technical quality and the results are conclusive and convincing. This reviewer does not have any specific suggestions on how to improve the experiments and/or data analysis.

Authors: We thank the Reviewer for appreciating the quality of our work.

My enthusiasm is a bit dampened by the lack of connection between the possible roles of NR2F1 in cortical gyrification based on human MRI data and in regulating progenitor proliferation based on mouse knockout data. Whether these two functions are linked remains to be demonstrated. The authors present possible ways to proceed in the discussion of the manuscript.

Authors: We fully agree with the Reviewer that further studies will be necessary to strengthen the link between NR2F1-dependent neurogenesis and patient cortical misfolding. To this aim, we are working on human organoids and try to implement protocols that can mimic brain folding. We also plan to challenge NR2F1 role in gyrencephalic models, such as the ferret. We have nevertheless down toned this possible link in the Discussion.

2nd Editorial Decision

24th Mar 2020

Thank you for submitting a revised version of your manuscript. Your study has now been seen by two of the original referees, who find that their main concerns have been addressed and are now in favour of publication of the manuscript. There now remain only a couple of minor editorial issues that have to be addressed before I can extend formal acceptance of the manuscript.

REFeree REPORTS:

Referee #1:

The authors have addressed most of my concerns. Most importantly, they included all the Stats and added data from heterozygous mice. They do leave some comments unaddressed (such as variant pathogenicity of patient 5 and defects in anterior brain areas) but promise that these features are in progress and will be published elsewhere.

Referee #2:

The authors have satisfactorily addressed all my previous concerns. But one new minor issue came out in Figure 6 as a result of the response to my last minor concern (indication of angle of mitotic cleavage plane). In addition to indicating the angle values, as requested, the authors have included schematic drawings to symbolize daughter cells resulting from the different division planes. This is very nice except that ideally the drawings should have a better correspondence with the appearance of those cell types in real tissue, which is specially noteworthy for horizontal division planes. This is a cosmetic point, but in my opinion worth paying attention to maintain the very high standards of this fantastic study.

2nd Revision - authors' response

1st Apr 2020

The authors performed the requested editorial changes.

Corresponding Author Name: Michèle Studer

Manuscript Number: EMBOJ-2019-104163R